# Impact of Financial Development Shocks on Renewable Energy Consumption in Saudi Arabia

**Raga M. Elzaki** [1,2]

1  Department of Agribusiness and Consumer Science, College of Agriculture and Food Science, King Faisal University, Al-Ahsa 31982, Saudi Arabia; rmali@kfu.edu.sa
2  Department of Rural Economics and Development, Faculty of Animal Production, University of Gezira, Wad Medani 20, Sudan

**Abstract:** The demand for renewable energy is increasing globally due to concerns about climate change, pollution, and the finite nature of fossil-fuel resources, and renewable energy has been recognized as a significant factor in realizing sustainable development. The government of Saudi Arabia adopted the reduction in fossil-fuel subsidies policy as a financial motivation for supporting both the production and consumption of fossil fuels. Therefore, this study aims to investigate the influence and shocks of Saudi financial development indicators on renewable energy consumption (REC) and to examine the track of causality between financial development indicators and REC. The study covers the annual data period of 1990–2021 and applies the Basic Vector Autoregressive model (VAR), the Granger causality test, forecast-error variance decomposition (FEVD), and the impulse response function (IRF). In the short run, the VAR results indicate a positive and significant impact of stock price volatility and private credit on REC. The results of causality between REC and financial development indicators were conflicting. The Granger causality test shows significant causality running from stock price volatility and private credit to REC. The FEVD results reveal that REC variation is explained by its innovative shocks and has a positive response to shocks in financial development. The IRF results show that REC has a positive response to shock on private credit, liquid liabilities, and stock price volatility. Authorities can encourage investment in renewable energy consumption by providing financial incentives; also, governments can foster national and international partnerships between investors, policymakers, and industry stakeholders. Employing different determinants of financial development indicators and incorporating population factors in the REC function will be highly recommended for forming the renewable energy demand in Saudi Arabia. Conducting a micro-level analysis of specific sectors within renewable energy, such as solar, wind, and others, can provide actionable insights for policymakers.

**Keywords:** renewable energy; financial development; VAR; Saudi Arabia

## 1. Introduction

Globally, in the last decades, renewable energies have been a major focus of investment, specifically solar photovoltaics and wind, and now they account for more than 80% of total investment in renewable energies globally [1]. The global investment in clean energy was estimated at USD 1.6 trillion in 2022. On average, USD 339 billion per year was committed globally for renewable power generation, compared to USD 135 billion for fossil-fuel power generation [2]. More than 60% of investment in renewables is derived from the private sector [3]. The demand for renewable energy usage has been increasing across the world, as renewable energy has been recognized as a significant factor in realizing sustainable development [4]. The last decades have been characterized by global crises, involving food, finance, and energy prices, which are linked to disastrous climate change [5]. In the context of the global financial crisis, financial development can exert both positive and negative consequences on the economy. On the one hand, it can improve economic growth

by providing capital for investment and facilitating the efficient allocation of resources [6]. On a similar line, financial development can raise access to financial services for low-income families and promote entrepreneurship by reducing poverty [7]. Likewise, financial development can also contribute to financial instability and systemic risk if financial institutions and markets are not properly controlled and regulated; further, this can result in financial crises and economic deterioration [8,9].

Meanwhile, understanding the role of financial development is crucial for renewable energy consumption for several reasons because financial development can provide incentives for the adoption of renewable energy, such as tax credits, subsidies, and other financial incentives [10,11]. These incentives can assist in decreasing the cost of renewable energy projects and make them more competitive with traditional energy sources [12]. In addition, Ref. [13] stated that the initial cost of installation is considered the key barrier to the approval of renewable energy. Financial development can afford access to capital and financing opportunities that can help overcome this barrier and increase investment in renewable energy projects [14]; also, renewable energy projects can involve significant risks, such as technological risk [15]. Financial development can provide risk management tools, such as insurance and hedging products [16], which can help mitigate these risks and encourage investment in renewable energy projects.

*Financial Development and Renewable Energy Consumption in Saudi Arabia*

In the context of Saudi Arabia, over the last years, the government of Saudi Arabia has adopted the reduction in fossil-fuel subsidies policy as a financial motivation for supporting both the production and consumption of fossil fuels, oil, coal, and gas. As a further target of this policy, the country has been making efforts to enhance its utilization of renewable energy sources, particularly solar and wind energy, by reducing its requirement on fossil fuels and reducing its subsidies. The forecasting energy demand in Saudi Arabia can be reduced by 5–10% [17]. Figure 1a displays the trend of subsidy reduction in oil, electricity, and gas. Recently, Saudi Arabia has aimed at producing 50% of its electrical energy from renewable sources by 2030, which includes a mix of wind, solar, and other sources [18]. Across 2021, the contribution of the final REC by sector accounted for 66%, 31%, 2%, and 1% for residential, commercial, industrial and, other sectors, respectively, while the contribution of the final REC by technology accounted for 84%, 9%, 5%, 1%, and <1% for charcoal, concentrated solar power, solid biofuels, solar photovoltaics, and wind, respectively [3].

Saudi Arabia is still in the early stages of its renewable energy shift, and the country has made significant progress in recent years. The government's guarantee to diversify its energy mix and the constructive renewable energy resources available in the country offer a strong foundation for further growth in renewable energy consumption.

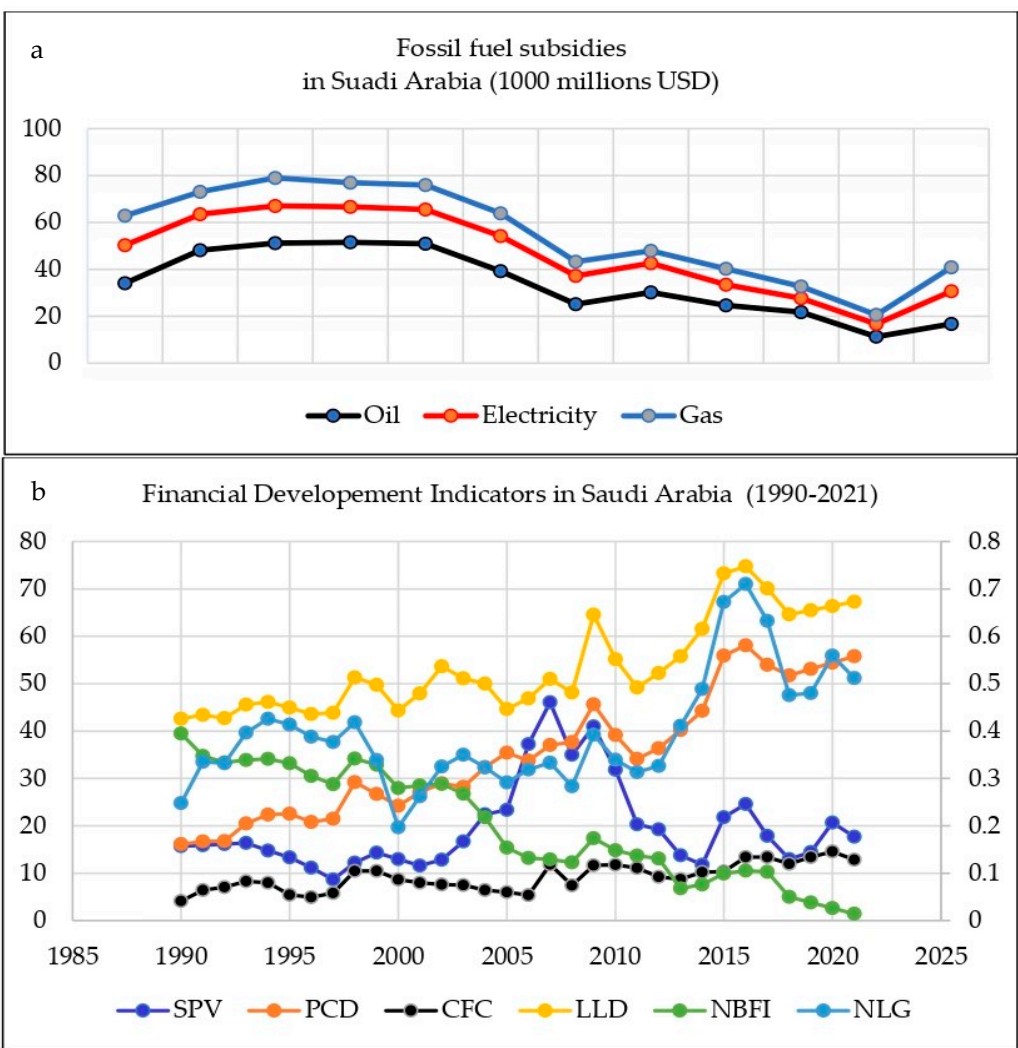

**Figure 1.** (**a**) Source: [19] and author's design (2023). (**b**) Source: [20] and author's design (2023). Note: (1) The right axis represents, SPV, PCD, CFC, LLD, and NBFI values. (2) The left axis represents NLG values.

To raise and boost the renewables share in Saudi Arabia, the Ministry of Energy in Saudi Arabia launched the National Renewable Energy Program (NREP) in 2019, intending to generate 27.3 GW of renewable energy by 2024 and up to 60 GW by 2030 [18]. The program aims to develop solar, wind, and other renewable energy projects and work on raising the renewable energy sector by establishing a competitive national market that contributes to the development of private-sector investments and promotes a combination of public- and private-sector investments [21].

Generally, the financial sector in Saudi Arabia is dominated by the banks; the Saudi Central Bank (SCB) is a regulator of the financial sector. In recent years, Saudi Arabia has made significant progress in financial development and has undergone significant reforms to become more advanced and integrated with the global financial system. The Financial Sector Development Program's partners have made continuous and boosted attempts to keep in step with the main transformations in the Kingdom since the launch of Vision 2030 [22]. The SCB has launched several initiatives planned to promote financial inclusion, involving the establishment of a credit bureau and the introduction of regulations to promote micro-finance. However, despite these developments, there are still challenges facing the financial sector in Saudi Arabia, including the need for further reforms to improve the regulatory framework, enhance corporate governance, and promote competition in

the sector [22]. In addition, the financial sector system has defeated many challenges considering the consequences of the COVID-19 pandemic.

The trend of the annual data of some important financial development indicators, namely, stock price volatility (SPV), private credit by deposit money banks to GDP (PCD) (in %), consolidated foreign claims of BIS reporting banks to GDP (CFC), liquid liabilities to GDP (LLD) (in %), non-bank financial institutions' assets to GDP (NBFI) (in %), and non-life insurance premium volume to GDP (NLG) (in %), can be seen in Figure 1b. We found that most financial development indicators have been showing steadily declining trends in recent years, with relatively fluctuating trends across the period 1990–2021.

Despite the reduction in carbon dioxide ($CO_2$) emission in the last decades, which accounted for 15.1, 14. 7, and 14.3 tons per capita, in 2018, 2019, and 2020, respectively, Saudi Arabia still faces serious issues from fuel ignition [23]; therefore, Ref. [21] worked on advancing the renewable energy sector by creating a competitive local market that contributes to the development of private-sector investments and promotes partnerships between the public and private sectors. However, Saudi Arabia is still heavily reliant on fossil fuels, particularly oil, for its energy needs. However, the government's drive towards renewable energy is viewed as a step toward diversifying the country's energy mix and lowering its carbon footprint. For studying the impact of financial development indicators on REC, the following research questions are raised: Do financial development indicators expand REC in Saudi Arabia? What is the causal relationship between REC and financial development? Then, in relation to these problems and questions, this paper aims to explore the influence and shocks of Saudi's financial development indicators on REC and to establish the direction of causality between financial development indicators and REC.

The contribution of this paper to the recent literature is threefold: first, it is based on BVAR forecasting for testing a theoretical linkage between financial development indicators and REC. Second, the generalizability of the literature review is enriched through more appropriate outcomes of the two concepts (financial development indicators and REC), and researchers can gain a better understanding of the mechanisms through which financial development can contribute to the progress of the renewable energy sector. Third, the empirical results may be more reliable for policymakers, providing them with further comprehensive knowledge to plan policies. The innovation/novelty of this study can help policymakers suggest effective and supportive financial policies that will attract investment and accelerate the positioning of renewable energy technologies.

The study is organized as follows. An introduction has been given in Section 1. Section 2 consists of a review of empirical studies related to financial development and REC concepts. Section 3 presents the data, variable descriptions, and methodological framework. Empirical results are presented in Section 4, and Section 5 concludes the study.

## 2. Review of Empirical Studies

Globally, since REC has increased continuously, the nexus between REC and financial development has garnered significant attention from researchers and policymakers alike over the past decades [4,24]. Scholars have identified the key variables and mechanisms that link financial shocks to renewable energy investments, such as capital availability, market dynamics, policy uncertainty, and investor behavior [25–28]. The relationship between REC and financial development has been examined by several investigators using different datasets and applying different mathematical and econometric methods in dissimilar regions. One study investigates the relationship between REC and the financial development index in Nigeria, utilizes time-series data, and uses financial institutions and financial markets indicators by applying the fixed effects model, finding that financial development is significant for renewable energy consumption [29]. A similar study considers the impact of financial development indexes using mixes of econometrics models, fully modified ordinary least square (FMOLS), dynamic ordinary least squares (DOLS), canonical cointegrating regression (CCR), Bayer and Hanck cointegration, and frequency-domain causality

tests for investigating the long-run interaction among the impact of financial development indexes on REC and environmental sustainability from a global perspective. The results show that financial development negatively influences CO2 emissions. Also, renewable energy usage boosts environmental quality in the world [30].

Another study conducted in the USA using the novel Fourier causality test with wavelet transforms found that financial development encourages renewable energy consumption at high quantiles in the medium and long run [31], while [32] applied ARDL co-integration, which indicated that in the long run, financial sector intermediation had a significant positive effect on energy demand in the Nigerian economy. Furthermore, another study [33] evaluated the relationship between REC and financial development, employing panel non-linear Autoregressive Distributed Lag (ARDL), and found that non-linear estimation supports the long-run asymmetric relationships between financial development, trade openness, capital flows, and renewable energy consumption; also, under the vector error correction estimation (VECM), they observed a long-run causality of financial development for REC [33].

On the other hand, Ref. [34] investigates the long-run effect of the financial development level of developing countries on renewable energy consumption using the FMOLS approach. The observed findings indicate the existence of a long-run connection between renewable energy consumption and financial development; besides, financial development increases the demand for renewable energy. In the same manner, Ref. [35] examined the relationship between energy consumption, financial development, and economic growth in Azerbaijan, employing mixed cointegration techniques (Johansen tests, Pesaran's bounds test, and the Gregory–Hansen test) for time-series data. The Johansen and Pesaran's bounds test showed the existence of a significant change relationship. In contrast, the Gregory–Hansen test results showed no statistically significant change in the long-run relationship.

Most of the above-mentioned studies have confirmed the appositive relation between financial development and REC; however, some studies of REC and the financial development nexus did not produce consistent findings; for instance, Ref. [36] investigated the impact of financial development and economic growth on REC in India using annual data and performed a DOLS model and a Granger causality test under a VECM model environment. This study argued for significant and positive influences of economic growth and financial development on renewable energy consumption. In contrast, a similar study performed in China [37] used a combination of ARDL, a pooled mean group (PMG) model, and Granger causality based on panel data and found that, in the long run, economic growth stimulates REC, whereas financial development negatively affects REC. But in the short run, an inverse result was noted: financial development has a positive effect on REC, while economic growth negatively affects REC. Also, the study observed unidirectional causal relationships between financial development and REC [37]. Compared with other panel data methods, Ref. [38] used VECM and the Granger causality test to explore the relationship between REC and foreign direct investment. Their empirical results indicate that there is a long-term and stable equilibrium relationship between foreign direct investment and renewable energy consumption; however, in the short term, foreign direct investment does not significantly cause renewable energy consumption.

A study performed in Asia Pacific Economic Cooperation (APEC) countries, Ref. [39] investigated the long-run relationship between the financial development index, REC, and the environment, applying econometric approaches, namely, feasible generalized least squares (FGLS), Augmented Mean Groups (AMG), and Correlated Effect Mean Groups (CCEMG); the outcomes reveal that financial development and renewable energy consumption significantly accelerate environmental quality [39]. Based on a system, the Generalized Method of Moments (GMM) estimator, Ref. [40] found that financial development had positive influences on REC in emerging economies, while the authors of [41] conducted a study in MENA countries which tested the environmental Kuznets curve (EKC) and indicated that financial development has adverse and significant effects on environmental degradation and affirmed the legitimacy of the EKC hypothesis in these countries. Also, Ref. [42]

indicated that green hydrogen is an important sustainable clean fuel, and its utilization is essential for both environmental preservation and energy security in MENA countries.

In the context of Saudi Arabia, most studies search for the effect of renewable energy on ecological footprints, carbon dioxide emissions, economic growth, and renewable energy systems and types [43–46]. Studies examining the connection between renewable energy consumption and financial development are limited in Saudi Arabia. One study examines the causal relationship between renewable energy consumption and financial development with real GDP and trade in the Gulf Cooperation Council (GCC) countries, employing a multivariate Granger causality and panel error correction model (ECM), which indicates no evidence of causality in the short run between exports and REC. However, a negative impact of financial development on economic growth was observed [47]. Another study conducted in Saudi Arabia investigated the impact of financial development factors (using real domestic credit in the private sector and real capital use) on total energy consumption, using the ARDL model. It was found that, in the long run, financial development improves energy demand in Saudi Arabia [48]. A further study conducted in Saudi Arabia using a causality test confirmed that green growth slows the impacts of financial development and trade globalization [49]. Another study using Non-Linear Autoregressive Distributed Lag (NARDL) found that, in the long run, a positive shock in energy consumption and negative shocks in financial development stream $CO_2$ emissions in Saudi Arabia [50].

While considering these studies, it can be observed that different econometric approaches, such as the vector error correction model (VECM), ARDL bounds testing, ordinary least squares (DOLS), Granger causality, the Generalized Method of Moments (GMM), etc., were used in these investigations. Therefore, a few studies investigated the impact of financial development on REC by applying vector autoregressive models (VARs) [51] and used time-series data applying VARs to investigate how much financial development indicators (stock market development, credit market growth, and the growth of international investment) have contributed to the growth of renewable energy in China and found that the financial sector contributes significantly to shifting the structure of energy in China. In addition, the authors of [14] performed a study in the European Union using a GMM panel VAR and found that the banking sector, the bond market, and the capital market have a positive effect on the share of renewable energy consumption.

In conclusion, the contradictory findings obtained from the mentioned studies are generated by the periods and variables selected, the different econometric techniques [52], and the different zones [53]. From the cited literature review, some gaps were observed: Firstly, studies applying the Basic VAR model for investigation of the connection between financial development indicators and REC are neglectable. Secondly, in comparing our study with previous studies conducted in Saudi Arabia, it was found that no studies have been carried out in Saudi Arabia to examine the relationship between the financial development indicators (such as stock price volatility, private credit by deposit money banks to GDP, and liquid liabilities to GDP) and total renewable energy consumption. Hence, this paper increases the existing literature to fill the observed gaps.

## 3. Materials and Methods

### 3.1. Data and Descriptive Statistics

The current study examined the impact of financial development indicators on renewable energy consumption in Saudi Arabia. We chose Saudi Arabia as a case study given the justification that it is one of the wealthiest countries in the world in terms of total GDP [46,47]: the total GDP was estimated at USD 1,010,589.333 million in 2022, which represented an increase of 12% in contrast to 2021; therefore, the country can invest to the utmost in renewable energy. Due to data viability, the annual time series (1990–2021) have been collected from the World Bank, the Sustainable Energy for All database [22], the World Bank, the global financial development database [20,48]. The selected variables involve total renewable energy consumption (REC) in terajoules (TJ) as a proxy of sustainable development factors (including solar, wind, hydropower, biofuels, and others). Stock price

volatility (SPV), private credit by deposit money banks to GDP (PCD) (in %), and liquid liabilities to GDP (LLD) (in %) were selected as financial development indicators (FDIs).

In this study, the reason for choosing the REC variable is that Saudi Arabia is greatly dependent on fossil fuels, particularly oil, for its energy needs. The reason for choosing the financial development indicators is related to their significance for economic growth, attracting foreign investment, and generating income through renewable energy consumption, which can play a vital role in investment in renewable energy technology in Saudi Arabia. Furthermore, the advanced financial system in Saudi Arabia can sustain credits to the renewable energy industry in an effective way, since REC requires high startup costs, and long-term debt repayment [31]. The variable definitions and the descriptive statistics of the selected variables are provided in Table 1.

**Table 1.** Descriptive statistics of the selected variables.

| Variable | Mean | Std. Dev. | Min | Max | Obs. |
|---|---|---|---|---|---|
| 1 * REC | 263.22 | 100.28 | 161.90 | 536.58 | 32 |
| SPV | 19.52 | 9.21 | 8.68 | 46.06 | 32 |
| 2 ** PCD | 35.02 | 13.09 | 16.11 | 58.11 | 32 |
| 3 ** LLD | 53.50 | 9.81 | 42.60 | 74.73 | 32 |

Note: REC: Renewable energy consumption (TJ) is the total final energy consumption; SPV: Stock price volatility refers to the average of the one-year volatility of the national stock market index; PCD: Private credit by deposit money banks to GDP (%) is the financial resources provided to the private sector by domestic money banks as a share of GDP; LLD: Liquid liabilities to GDP (%) is the ratio of liquid liabilities to GDP. Sources: * = data derived from [20,54]; ** = data derived from [20]. 1. This indicator is derived from energy-balance statistics and is equivalent to total final consumption, ignoring non-energy use [22]. 2. Domestic money banks comprise commercial banks and other financial institutions that accept transferable deposits, such as demand deposits [20]. 3. Liquid liabilities are also known as broad money, which is recognized as M3. M3 = deposits in the central bank (M0) + transferable deposits and electronic currency (M1) + time and savings deposits, foreign currency transferable deposits, certificates of deposit, and securities repurchase agreements (M2) + travelers' checks, foreign currency time deposits, commercial paper, and shares of mutual funds or market funds held by residents [20].

### 3.2. Econometric Methods

As an initial step in the present study and before implementing the study models, some relevant pre-test analyses, such as normality and unit roots, were analyzed.

### 3.2.1. Unit Root Test

For testing the stationarity level of the time-series variables, first, we applied the proposed developed version of [55], which is employed by [56], based on generalized least squares (GLS) detrended data, $\Delta ydt$. We applied the Ng-perron test because it is more suitable than the traditional tests [57] and it is also more efficient for large negative errors than the PP test [58]. We analyzed the properties of four Ng-Perron tests involving modifications of the subsequent four-unit root tests: Phillips–Perron $Z\alpha$ and $Zt$, Bhargava R1, and ERS, which is considered a feasible optimal point test, collectively referred to as the M tests. These properties tests have the following formulas:

$$MZ_\alpha^d = \left( T^{-1} \left( y_T^d \right)^2 - f_0 \right) / 2k \tag{1}$$

$$MSB^d = (k/f_0)^{\frac{1}{2}} \tag{2}$$

$$MZ_t^d = MZ_\alpha^d \times MSB^d \tag{3}$$

$$MPT_T^d = ((\bar{c})^2 k + (1 - \bar{c}) \, T^{-1}) \left( y_T^d \right)^2 / f_0 \tag{4}$$

where the statistics $MZ_\alpha^d$ and $MZ_t^d$ are efficient versions of the PP test and:

$$k = \sum_{t-2}^{T} \left( y_{t-1}^d \right)^2 / T^2 \tag{5}$$

$$f_0 = \sum_{j=-(T-1)}^{T-1} \varnothing(j).k(j/l) \tag{6}$$

where $\bar{c} = -13.5$; $l$ is a bandwidth parameter (which acts as a truncation lag in the covariance weighting); $\varnothing(j)$ is the jth sample autocovariance of residuals; $t = 1,2,3.\dots\dots, T$ represents an index of time; Z is a set of deterministic components; $\alpha$ is the likelihood ratio statistic; and $d$ is the diagonality of the matrix.

Second, we used the Zivot–Andrews unit root test proposed by [59] to capture a single structural break in the time-series data. The Zivot–Andrews test not only tests the unit root properties of each variable but also considers one structural break. The test in [59] applied the sequential Augmented Dickey–Fuller (ADF) [60] test to find the break corresponding to the A, B, and C models and used the equations formed as follows:

$$\Delta Y_t = K + \alpha Y_{t-j} + \beta_t + \gamma_i UD_t + \sum_{j=1}^{k} d_j \Delta Y_{t-j} + \varepsilon_i \quad \text{Model(A)} \tag{7}$$

$$\Delta Y_t = K + \alpha Y_{t-j} + \beta_t + \theta UD_t + \sum_{j=1}^{k} d_j \Delta Y_{t-j} + \varepsilon_i \quad \text{Model(B)} \tag{8}$$

$$\Delta Y_t = K + \alpha Y_{t-j} + \beta_t + \theta UD_t + \gamma_i DT_t + \sum_{j=1}^{k} d_j \Delta Y_{t-j} + \varepsilon_i \quad \text{Model(C)} \tag{9}$$

where $\Delta$ is the first difference and $Y_t$ denotes a variable series containing the unit roots referred to the existing study, REC, SPV, PCD, and LLD. The $Y_{t-j}$ terms on the right-hand side of the three equations allow serial correlation and prove that the disturbance term is white noise with variance $\sigma 2$. $UD_t$ is an indicator dummy variable for a mean shift appearing at each possible time break date (TBD), while $DT_t$ is the corresponding trend variable, whereas:

$$UD_t = \begin{cases} 1 & if\ t > TBD \\ 0 & otherwise \end{cases} \tag{10}$$

And:

$$TD_t = \begin{cases} 1 - TBD & if\ t > TBD \\ 0 & otherwise \end{cases} indci \tag{11}$$

The null hypothesis of the three models is $\alpha = 0$, which implies the existence of a unit root in a series $(Y_t)$ with drift that rejects any structural break, whereas the alternative hypothesis, $\alpha < 0$, implies that the series is trend stationary. Model A permits a change in the intercept of a series and Model B permits a change in the trend of a series, while Model C permits changes in both intercepts and trends. Most scholars [61,62] have applied Model A and/or C. In this study, Model A was applied for the analysis of unit roots because it is more comprehensive than Model B, as it allows for a break in intercepts.

### 3.2.2. The Basic VAR Model

After testing the unit root problem in the time-series variables, the multivariate Basic VAR approach was used. The reasoning for using this method was as follows: it is suitable for analyzing the dynamic interactions between multiple variables over time, it allows the possibility of both REC and financial development being endogenous, and it can capture the dynamic relationships between multiple variables, making them a flexible tool for analyzing complex systems. Second, Basic VAR models can be used for forecasting,

allowing us to make predictions about future values of the variables in the system and examine the causality [63].

The VAR model was principally suggested by [64] and recently has been broadly applied in macro-economic analysis of energy and financial development [65–67]. Given the M times-series variables, $Y_t = (Y_{1t}, \ldots \ldots, Y_{Mt})$, following [68], the Basic VAR model takes a reduced simultaneous form as follows:

$$Y_t = v + A_1 y_{t-1} + \cdots + A_\rho y_{t-\rho} + u_t = v + AY_{t-1}^{t-\rho} + u_t \tag{12}$$

where $Y_t$ is the vector of endogenous variables, represented in our study as REC, SPV, PCD, and LLD, which are being forecasted, and the only deterministic component is a constant term denoted by $v$, which is a constant term ($M \times 1$), a vector of the intercept. $A$ is the matrix of coefficients for the i$^{th}$ lag ($M \times n$) polynomial matrix in the backshift operator with lag length p, and $u_t$ ($n \times 1$) is the vector of white-noise error terms, i.e., the vector comprising the reduced-form residuals, which in general will have non-zero correlations. Using Equation (12) for a given VAR order, $p$, an estimation can be conveniently performed by equation-wise ordinary least squares (OLS) including 2 lags. We applied the lag-length selection criteria for selecting the number of lags according to an explicit statistical information criterion.

The Lag-Length Selection Criteria

After we performed the Basic VAR analysis, the selection of the lag length was essential for determining the lag length for the VAR(p) model using the optimum model selection criteria. We utilized the lag-length selection criteria to determine the appropriate lag length according to [69], which involves the Akaike Information Criterion (*AIC*), the Schwartz–Bayesian (*SBIC*) criterion, the Hannan–Quinn (*HQIC*) criteria, likelihood ratios, the sequential modified (LR) criteria, and Final Prediction Error (FPE). The following are the formulas for each lag-length criterion:

$$AICp = -\frac{n}{2(1 + log2\pi)} - \frac{n}{2log\delta^2} - p \tag{13}$$

$$SBICp = log(\delta^2) + \left(\frac{logn}{n}\right)p \tag{14}$$

$$HQIC = log\delta + \left(\frac{2loglogn}{n}\right)p \tag{15}$$

$$LR = n(log[\Sigma P] - log[\Sigma P]) \tag{16}$$

$$FPE = n[(n + p)(n - p)log[\Sigma P]) \tag{17}$$

where $\delta^2$ represents the maximum-likelihood (ML) estimator of the variance of the regression disturbances, $\sum p$ represents the estimated sum of squared residuals, $n$ is the number of estimated parameters, and $p = 0, 1, 2 \ldots \ldots P$, where P is the optimum order of the model selected. The HQ and SC criteria are both consistent [70], that is, under general conditions, the order considered with these criteria converges in probability or almost surely to the true VAR order $p$ if pmax is at least as large as the true lag order [71]. We approved the model selection fitting to the lowest *AIC* or *SBIC* value.

Granger Causality Test

The next stage of analysis in this study was Granger causality testing in the VAR environment. We used Granger causality tests because they provide evidence of the direction and strength of causality between financial development and renewable energy and help to establish whether financial development drives renewable energy development or vice

versa. The Granger causality test proposed by [72] can be tested in a VAR multivariate model to test for the simultaneousness of all integrated variables [73]. We suggest the Granger causality test for the case of LogFDI and LogREC, which are involved as a first step in the estimation of the following VAR models:

$$logFDI_t = \alpha_1 + \sum_{i=1}^{n} \beta_i logREC_{t-1} + \sum_{j=1}^{m} \delta_j logFDI_{t-j} + \mu_t \qquad (18)$$

$$logREC_t = \theta + \sum_{i=1}^{n} \varnothing_i logFDI_{t-1} + \sum_{j=1}^{m} \varphi_j logREC_{t-j} + \omega_t \qquad (19)$$

where *FDIs* (financial development indicators) could be SPV, PCD, or LLD; $\alpha$ and $\theta$ are the intercepts of the two equations, respectively; $\beta_i$ and $\varnothing_i$ represent the coefficients of the equations; and $\mu_t$ and $\omega_t$ are error terms for the two equations, respectively. The symbols m and n represent the maximum number of lags for each of the variables.

Based on the estimated OLS coefficients for Equations (18) and (19) and following [74], four different hypotheses about the relationship between REC and financial development indicators can be clarified:

1. Unidirectional Granger causality from FDIs to REC. In this condition, FDIs increase the prediction of REC but not vice versa.

2. Unidirectional Granger causality from REC to FDIs. In this condition, REC increases the prediction of FDIs but not vice versa.

3. Bidirectional Granger causality from FDIs to REC. In this condition, FDIs increase the prediction of REC and vice versa.

4. Independence between FDIs and REC. In this condition, there is no Granger causality in any direction.

Impulse Response Functions and Forecast-Error Variance Decomposition Tests

For examining the dynamics of the VAR model for estimating the progress of variable shocks, we focused on impulse response functions (IRFs) and forecast-error variance decompositions (FEVDs). VAR analysis frequently involves the estimation of IRFs and FEVDs, which are the fundamental elements of the VAR method. Finally, we followed [75,76] in setting the IRFs and FEVDs for a 10-year forecast horizon (h). The IRFs were chosen because IRF analysis allows for the assessment of how financial development shocks or renewable energy shocks propagate through a system, providing valuable insights into the dynamic interactions between these variables.

The orthogonalized impulse response function was employed to evaluate the sensitivity of the dependent variable to changes (shocks) in each of the variables, i.e., the shock of financial development indicators on REC. The impulse response at horizon h of the variables to an exogenous shock to variable y can be easily displayed with Cholesky decomposition proposed by [64], as follows:

$$y_t = \sum_{i=0}^{\infty} \vartheta_i \, \nu_{t-i} [\vartheta_0 = I_k is \ the \ (K \times K) \ identity \ matrix] \qquad (20)$$

$$\vartheta_i = \sum_{j=1}^{i} \vartheta_{i-j} A_j [i = 1, 2, 3, \ldots \ldots] \qquad (21)$$

where $\vartheta_i$ values are impulse responses of the model; $Aj = 0$ for $j > p$ (for a *k*-dimensional VAR (p) process); and $\nu_t$ represents the orthogonal residuals [63]. The IRFs do not imply causation, but they clarify the probability of a shock on one variable affecting the other variables [77]. Additionally, the decomposition is not exclusive but is affected by the ordering of the variables [78]. Variance decomposition provides a rationale for the percentage of change in the dependent variable explained by its shocks, and it is used to forecast exogenous shocks of the variables [79].

Follows [78], the h-step-ahead predictor vector error equation used in this study is written as:

$$Y_{it+h} = E[Y_{it+h}] = \sum_{M=0}^{h-1} A_j [e_{i(t+h-i)} \vartheta_i \quad (22)$$

where $Y_{it+h}$ is the observed vector at time $t + h$; $E[Y_{it+h}]$ is the h-step-ahead forecast vector error made at time $t$ or the orthogonalized shock $E[Y_{it+h}]$ is the h-step-ahead predictor, which is the g-step-ahead predictor vector made at time $t$; and the orthogonalized shocks $e_{it}M^{-1}$ (with M matrix) have a covariance matrix, $I_M$. The FEVD model was chosen because it can help identify the relative importance of shocks from financial development indicators and renewable energy consumption in explaining the forecast-error variance of each variable.

In overall VAR models, Granger causality tests, FEVD, and IRF analysis were chosen for their ability to capture the dynamic relationships, causality, and interdependencies between financial development indicators and renewable energy consumption, providing a comprehensive understanding of their interactions over time.

## 4. Discussion of Outcomes and Results

In this section, we review the results of pre-test analyses and Basic VAR results based on the standard approaches defined in Section 3.

### 4.1. Preliminary Results

In Table 2, the selected variables for REC and SPV are positively skewed, with $p < 0.05$, which indicates that this variable is non-normally distributed and vice versa for PCD and LLD, which have a normal distribution. Further, the test of [80] confirms the results obtained from the skewness and kurtosis test. The results of the heteroskedasticity test [81] show that the variance error is not constant. To ensure the reliability and consistency of empirical results by reducing non-linearity or heteroscedasticity in the time-series dataset and for modeling purposes, all variables were transferred in logarithm form.

**Table 2.** Normality and residual diagnostic tests.

| Variable | Obs | Pr(Skewness) | Pr(Kurtosis) | Joint Test | | Normality Status |
| --- | --- | --- | --- | --- | --- | --- |
| | | | | Adj $X^2$ | Prob > $X^2$ | |
| REC | 32 | 0.0003 | 0.0316 | 13.46 *** | 0.00 | Non-normal |
| SPV | 32 | 0.0012 | 0.0740 | 10.97 *** | 0.00 | Non-normal |
| PCD | 32 | 0.3786 | 0.0487 | 4.68 | 0.10 | Normal |
| LLD | 32 | 0.0522 | 0.4191 | 4.50 | 0.11 | Normal |
| | JB Tests | | | Heteroskedasticity: Breusch–Pagan's Test | | |
| Variable | $X^2$ | Prob > $X^2$ | Normality status | $X^2$ | Prob > $X^2$ | Description |
| REC | 19.66 *** | (0.00) | Non-normal | 16.51 *** | (0.00) | Serial correlation |
| SPV | 13.58 *** | (0.00) | Non-normal | 4.85 *** | (0.02) | Serial correlation |
| PCD | 2.17 | (0.34) | Normal | 15.35 *** | (0.00) | Serial correlation |
| LLD | 3.89 | (0.14) | Normal | 10.34 *** | (0.00) | Serial correlation |

Note: *** = Levels of significance at 1%. $X^2$ = Pearson's chi-square tests. $H_0$: no serial correlation. Durbin–Watson d-statistic (7, 32) = 1.217352.

For examining cointegration between the selected variables, testing the stationarity of the selected variables is a crucial condition. For this, we applied the new unit root test of [56]. It is noted that the selected variable is non-stationary at levels. However, REC, SPV, and PCD take the stationary nature at the first difference, whereas LLD takes the stationary nature at the second difference with the intercept (Table 3). However, the Ng-

Perron unit root test has a limitation: the Ng-Perron test has limited power, i.e., it provides ambiguous and spurious results for some time-series data, even if unit roots exist. Also, the Ng-Perron test assumes that data are stationary over the entire time being analyzed, and it may not be able to detect unit roots if there are structural breaks originating from the series, which further disconfirms the stationarity hypothesis [48]. To overcome these limitations, we applied the unit root test of [59] with single structural breaks in intercepts to obtain more robust results. The structural break date (SBD) test considers the probability of exhibiting single structural breaks that are assumed to be endogenously determined. The findings from the Zivot–Andrews test for structural breaks and unit roots indicate that all the selected variables are initially identified as non-stationary. However, it is important to note that these variables become stationary after taking their first difference, despite the presence of structural break(s) found to be stationary. In the intercept condition result, the structural break dates were 1999, 2010, and 2014, observed in REC, SPV, and PCD, respectively. However, a significant break date was identified as 2009 for LLD, whereas in the trend condition result, the breaks were 2007 for REC, 2005 for SPV, 1997 for PCD, and 2016 for LLD.

**Table 3.** Unit root test results.

| Variable | Ng-Perron Test Statistics with Intercept | | | |
|---|---|---|---|---|
| | MZ$\alpha$ | MZt | MSB | MPT |
| LogREC (−1) | −14.9877 | −2.73746 | 0.18265 | 1.63477 |
| LogSPV (−1) | −51.3672 | −5.01774 | 0.09768 | 0.60084 |
| LogPCD (−1) | −36.6042 | −4.27765 | 0.11686 | 0.6706 |
| LogLLD (−2) | −29.0409 | −3.81041 | 0.13121 | 0.84415 |
| | Asymptotic Critical Values for Ng-Perron Test | | | |
| 1% | −13.8 | −2.58 | 0.174 | 1.78 |
| 5% | −8.1 | −1.98 | 0.233 | 3.17 |
| 10% | −5.7 | −1.62 | 0.275 | 4.45 |
| | Zivot–Andrews Unit Test Results | | | |
| | Intercept * | | Trend ** | |
| | t-Stat | SBD | t-Stat | SBD |
| LogREC | −9.380 | 1999 | −8.113 | 2007 |
| LogSPV | −6.571 | 2010 | −4.793 | 2005 |
| LogPCD | −6.076 | 2014 | −5.846 | 1997 |
| LogLLD | −5.261 | 2009 | −5.215 | 2016 |

* The critical values for the Zivot and Andrews test are −5.34, −4.80, and −4.58 at 1%, 5%, and 10% levels of significance, respectively. ** The critical values for the Zivot and Andrews test are −4.93, −4.42, and −4.11, at 1%, 5%, and 10% levels of significance, respectively. Source: Author's calculations (2023).

### 4.2. Estimation of the Basic VAR Model Results

The VAR model implies an equation for each variable describing its evolution with its lags and the lags of other variables; therefore, all the variables are symmetrically treated as endogenous. We estimated the VAR system using REC, SPV, PCD, and LLD as endogenous variables and the constant as an exogenous variable. In Table 4, the estimated model of REC indicates that REC reacts positively to a short-run change in SPV (−2) and PCD (−2) in Saudi Arabia.

**Table 4.** Basic VAR model results.

| Independent Variable | Dependent Variable (Equations) | | | |
| --- | --- | --- | --- | --- |
| | LogREC | LogSPV | LogPCD | LogLLD |
| LogREC (−1) | 0.471 [0.15] *** | 0.139 [0.176] | −0.029 [0.084] | 0.010 [0.064] |
| LogREC (−2) | −0.27 [0.14] * | −0.86 [0.172] | 0.043 [0.083] | 0.064 [0.062] |
| LogSPV (−1) | −0.119 [0.148] | 0.937 [0.174] *** | −0.116 [0.083] | −0.179 [0.063] *** |
| LogSPV (−2) | 0.441 [0.163] *** | −0.376[0.19] | 0.019 [0.091] | 0.060[0.069] |
| LogPCD (−1) | 0.579 [0.52] | 1.126 [0.611] * | 1.193 [0.292] *** | 0.443[0.221] |
| LogPCD (−2) | 1.15 [0.507] *** | −0.230 [0.59] | 0.018 [0.284] | 0.109[0.215] |
| LogLLD (−1) | 0.161 [0.65] | −1.676 [0.76] *** | −0.398 [0.362] | −0.32 [0.27] |
| LogLLD (−2) | 0.545 [0.59] | −0.147 [0.70] | −0.165 [0.336] | 0.279[0.255] |
| RMSFE | 0.088338 | 0.103461 | 0.049465 | 0.037482 |
| R-squared | 0.5706 | 0.7660 | 0.9260 | 0.8180 |
| Chi$^2$ | 39.86597 *** | 98.18493 *** | 375.264 *** | 134.8692 *** |

Note: Standard errors in square brackets. RMSFE: This means that the forecast errors (the differences between the predicted values and the actual values) are relatively small compared to the scale of the data. *** and * = levels of significance at 1%, and 10%, respectively. Source: Author's calculations (2023).

So, a unit increase in REC (−1), SPV (−2), and PCD (−2) causes REC to improve by 0.471, 0.441, and 1.15 per unit increase in REC, respectively, while a unit increase in REC (−2) causes REC to reduce by 0.27 units. A unit increase in SPV (−1) and PCD (−1) causes a significant increase in SPV by 0.937 and 1.126, respectively, and, vice versa, a unit increase in SPV (−2) and LLD (−1) causes a decrease in SPV by 0.376 and 1.676, respectively. This means that PCD (−1) positively impacts SPV, while LLD (−1) exerts a negative impact on SPV. This finding can be justified by the increase in liquid liabilities, which can generate and increase the risk associated with investing in renewable energy companies, which can in turn lead to higher stock price volatility and a decline in a company's stock price.

Also, a unit increase in PCD (−2) significantly increased REC by 1.15. Likewise, PCD (−1) had a significantly positive impact on PCD and LLD. Finally, SPV (−1) caused a significant reduction in LLD. This can justify the unwillingness of stockholders to invest in institutions with volatile stock prices, as stock price volatility reduces an institution's liquid assets. Volatility impacts option prices exponentially; hence, the dilutive effect of in-the-money options on blended earnings can affect growth estimates adversely, making volatile companies harder to hold [82].

Overall, the Root Mean Squared Forecast Errors (RMSFEs) of all the equations were dramatically lower (less than one), which indicates a better fit and more precise predictions of the selected variables. In comparing our results with other studies, we agreed with [83], indicating that the stock price volatility (national market stock) in the short term is significant with renewable energy. Moreover, Ref. [24] finds that stock market value affects renewable energy in the long run. Also, Ref. [84] finds that financial development promotes renewable energy use; however, Ref. [85] argues that REC does not react to a short-run change in bank-based financial development (stock price volatility).

### 4.3. VAR Diagnostic Results

We performed diagnostic tests on a VAR model to assess its reliability and validity in achieving accurate predictions about the connections between the selected variables. In this study, it was assumed that the optimal lag length for the BVAR model is one (1) because it has many more (three) stars (*) in pre-estimation and many more (four) stars in the post-estimation than the other lags, which will make it possible to employ the BVAR model. These criteria balance the goodness of fit of the model with the complexity of the lag

structure. Lower values of the criteria indicate a better trade-off between fit and complexity (Table 5).

**Table 5.** Optimal lag selection criteria.

| | **Pre-Estimation Lag Order Statistics** | | | | | | | |
| | **Sample: 1994 through 2021 Number of Obs = 28** | | | | | | | |
| Lag | LogL | LR | FPE | AIC | HQIC | SBIC | Df | *p*-Value |
|---|---|---|---|---|---|---|---|---|
| 0 | 113.886 | | $6.0 \times 10^{-9}$ | −7.57835 | −7.51928 | 7.38976 | | |
| 1 | 179.188 | 130.6 | $2.0 \times 10^{-10}$ * | −10.9785 | −10.6832 * | 10.0355 * | 16 | 0.000 |
| 2 | 192.636 | 26.895 | $2.6 \times 10^{-10}$ | −10.8025 | −10.2709 | 9.10516 | 16 | 0.043 |
| 3 | 214.397 | 43.522 * | $2.1 \times 10^{-10}$ | −11.1998 * | −10.4319 | 8.74809 | 16 | 0.000 |
| | Post-Estimation Lag Order Statistics | | | | | | | |
| | Sample: 1992 through 2021 Number of Obs = 30 | | | | | | | |
| Lag | LogL | LR | FPE | AIC | HQIC | SBIC | Df | *p*-Value |
| 0 | 115.395 | | | −7.36654 | −7.4263 | −7.36654 | −7.23948 | |
| 1 | 186.545 | 142.3 | $1.8 \times 10^{-10}$ * | −11.103 * | −10.8041 * | −10.1689 * | 16 | 0.000 |
| 2 | 200.051 | 27.012 * | $2.3 \times 10^{-10}$ | −10.9367 | −10.3988 | −9.25527 | 16 | 0.041 |

* Indicates lag order selected by the criterion (optimal lag), endogenous: exogenous: constant. LR: Likelihood ratio, sequential modified LR test statistic (each test at 5% level). FPE: Final prediction error. AIC: Akaike Information Criterion. HQIC: Hannan–Quinn Information Criterion. SBIC: Schwarz Information Criterion. Source: Author's calculations (2023).

The Wald lag exclusion test was used to examine the possibility of lag elimination of any variable in the VAR system. The Wald test is a safety test for the number of lags chosen from the selection criteria [86]. Based on the results of the Wald test shown in Table 6, we found that the selected variables used in the VAR were significant (*p*-value < 0.05). Thus, the VAR model will be estimated by using the lag in order number one, which is determined by the selection criteria in Table 5.

**Table 6.** Wald statistics and Lagrange multiplier test.

| Lag | **VAR Lag Exclusion Wald Tests for Equations** | | | | |
| | LogREC | LogSPV | LogPCD | LogLLD | All |
|---|---|---|---|---|---|
| 1 | 16.52674 (0.002) *** | 39.26219 (0.00) *** | 24.63254 (0.00) *** | 21.13896 (0.00) *** | 115.1125 (0.00) *** |
| 2 | 12.75615 (0.013) *** | 9.42758 (0.05) ** | 0.6563887 (0.96) | 3.509985 (0.78) | 32.61345 (0.00) *** |
| | Lagrange Multiplier Test | | | | |
| Lag | Chi$^2$ | Prob > Chi$^2$ | Df | Decision | |
| 1 | 23.9736 | (0.09) * | 16 | Accept | |
| 2 | 21.4786 | (0.16) | 16 | Accept | |

Ho: No autocorrelation at lag order. Note: The X$^2$ test statistic values are given, with the *p*-values in parentheses. ***, **, and * = levels of significance at 1%, 5%, and 10%, respectively. Source: Author's calculations (2023).

For greater model reliability, the study also applied the Lagrange multiplier (LM) test, which is a multivariate test statistic for autocorrelation in residuals up to the specified lag order. The null hypothesis of the LM test is the non-existence of serial correlation versus the alternative of autocorrelated residuals. Our results from Table 6 show that both the lag lengths accept the null hypothesis of no serial correlation, indicating that the error terms of the equations are not correlated, which suggests that the fitted VAR system is reasonable.

Further, we applied the VAR eigenvalue stability condition to check the stability of the VAR model. It is based on the eigenvalues of the matrix of coefficients in the VAR model (Equation (12)). In particular, the situation requires that all the eigenvalues of the coefficient matrix lie inside the unit circle in the complex plane, which means that they have a modulus or absolute value of less than one [87]. By examining Table 7, we can observe two specific eigenvalues: the first eigenvalue and the last eigenvalue. Notably, these eigenvalues have no imaginary part and are purely real, indicating that their imaginary part is zero (the eigenvalue lies entirely on the real number line). Consequently, the corresponding eigenvectors associated with these eigenvalues do not experience rotation or oscillation but instead undergo scaling or stretching along a specific direction. Table 7 and Figure 2 illustrate that no root lies outside the unit circle, as each modulus value in the table and the figure is lower than 1. This assessment implies that the VAR model fits the stability condition.

**Table 7.** The eigenvalue stability condition of the VAR model.

| Eigenvalue | Modulus |
| --- | --- |
| 0.9384261 | 0.938426 |
| 0.7592049 + 0.2719225i | 0.806433 |
| 0.7592049 − 0.2719225i | 0.806433 |
| 0.1220844 + 0.6400749i | 0.651614 |
| 0.1220844 − 0.6400749i | 0.651614 |
| 0.1028066 + 0.443515i | 0.455274 |
| 0.1028066 − 0.443515i | 0.455274 |
| 0.01691226 | 0.016912 |

Note: Eigenvalues are complex numbers with both real and imaginary parts. The left values represent the real values of the eigenvalues, while the right values represent the imaginary values (i), and the "+" and "−" signs indicate the signs of the imaginary parts of those eigenvalues. All the eigenvalues lie inside the unit circle, which proves that VAR satisfies the stability conditions. Source: Author's calculations (2023).

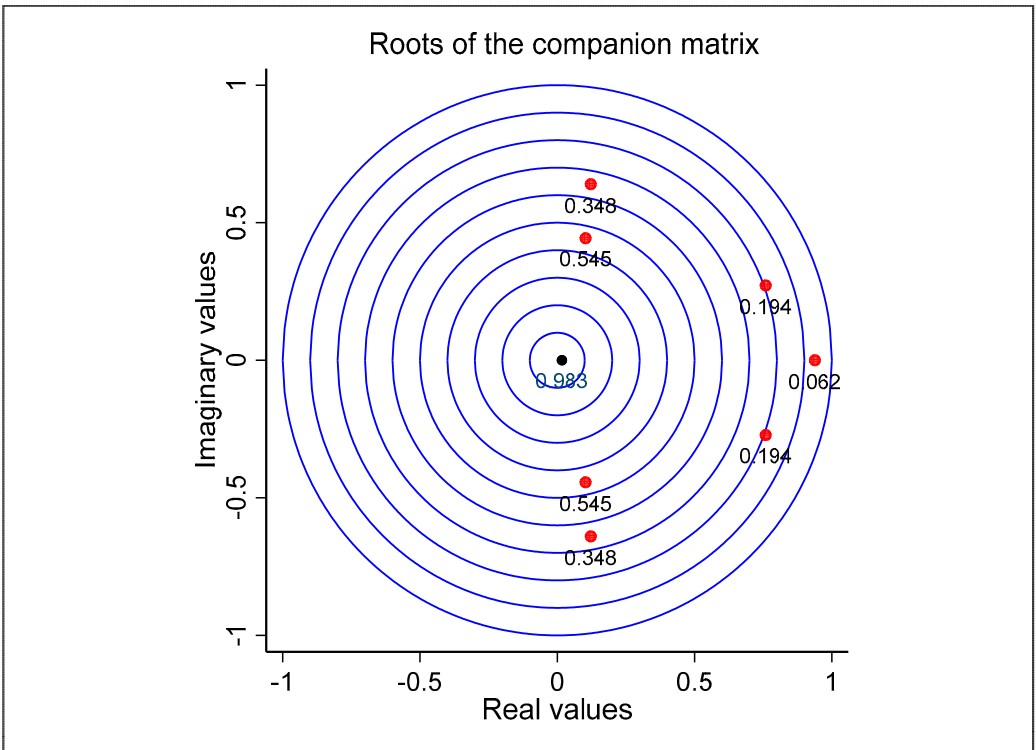

**Figure 2.** Note: All the values are less than 1. Source: author's design (2023).

### 4.4. Pairwise Granger Causality Approach for Robustness Checks

For investigating the possibility of causal relationships between logarithms for the time series of the selected variables and their direction, we used the Granger causality test in the VAR environment analysis. This is a useful approach that can assist in recognizing which variables are significant in our model and have a causal influence on other variables. All of the outcomes from the asymmetric Granger causality analysis are reported in Table 8. The results indicate that there is significant causality running from LogSPV and LogPCD to LogREC. The results also show significant causality running from LogPCD and LogLLD to LogSPV, which suggests that changes in credit availability and liquidity within the banking system can influence stock market dynamics. However, the results indicate that logREC, LogSPV, and LogLLD do not Granger-cause LogPCD, i.e., there are independence conditions between these variables. Meanwhile, LogREC is not sensitive to LogLLD, though LogSPV and LogPCD do Granger-cause LogLLD.

**Table 8.** Granger-causality Wald tests.

| Equation | Excluded | $X^2$ | Prob > $Chi^2$ | Results of Causality Run | Direction |
|---|---|---|---|---|---|
| LogREC | LogSPV | 10.533 | 0.005 *** | SPV→REC | Unidirectional |
|  | LogPCD | 6.906 | 0.032 ** | PCD→REC | Unidirectional |
|  | LogLLD | 1.6073 | 0.448 | No causality | Independent |
|  | ALL | 14.163 | 0.028 ** | REC←→FDI | Bidirectional |
| LogSPV | logREC | 0.63461 | 0.728 | No causality | Independent |
|  | LogPCD | 7.224 | 0.027 ** | PCD→SPV | Unidirectional |
|  | LogLLD | 7.6933 | 0.021 *** | LLD→SPV | Bidirectional |
|  | ALL | 11.343 | 0.078 | No causality | Independent |
| LogPCD | logREC | 0.28536 | 0.867 | No causality | Independent |
|  | LogSPV | 3.4615 | 0.177 | No causality | Independent |
|  | LogLLD | 2.8823 | 0.237 | No causality | Independent |
|  | ALL | 5.3268 | 0.503 | No causality | Independent |
| LogLLD | logREC | 1.7124 | 0.425 | No causality | Independent |
|  | LogSPV | 11.399 | 0.003 *** | SPV→LLD | Bidirectional |
|  | LogPCD | 12.677 | 0.002 *** | PCD→LLD | Unidirectional |
|  | ALL | 18.06 | 0.006 *** | LLD←→REC | Bidirectional |

Note: *** and ** = levels of significance at 1% and 5%; respectively.

Plausible justifications of the contradictory findings of an absence of causality include the presence of other variables that influence both the independent and dependent variables. For instance, factors like government policies, technological advancements, or macroeconomic conditions can simultaneously affect renewable energy consumption, stock price volatility, and financial indicators. These factors were not incorporated into the model because of data limitations.

From the findings, it is possible to draw a hypothesis that an increase in SPV and PCD leads to an increase in REC, i.e., there are unidirectional runs from SPV and PCD to REC. Also, there are bidirectional runs between SPV and PCD, and an increase in PCD leads to an increase in LLD, which implies unidirectional runs from PCD to LLD. From our Granger findings, we concluded that the results of causality between REC and financial development indicators were conflicting. We concluded from our Granger causality results that all four different hypotheses derived from Equations (18) and (19) exist. In comparing our results with other studies, Ref. [85] argues that renewable energy sources do not have a

statistically significant impact on financial development. However, Ref. [39] realizes the bidirectional causality relationship between financial development and REC.

### 4.5. Forecast-Error Variance Decomposition Results

Furthermore, in the present study, we analyzed FEVD using the Choleskey orthogonalization technique to detect the strength horizons beyond the selected time and chose 10 periods/horizons. We analyzed the FEVDs to quantify the extent to which forecast-error variance in a variable can be explained by innovations or impulses originating from that variable and the other variables in the system. Therefore, this approach estimates simultaneous shock effects. In this study, we took, for example, 3 years to represent the short run and 10 years to represent the long run. Table 9 shows that 65.69% of the variation in REC was caused by itself, while SPV and PCD caused increasing variation in REC, contributing 14.82% and 13.44%, respectively, in the last duration (10). At the same time, LLD caused a decrease in the variation of forecast errors in REC throughout the 10 years, ending with 6.15% in the last period (10). This indicates that REC is shocked by itself with larger percentages of forecast error than the selected financial development indicators of forecast error throughout the 10 years. The empirical evidence from Table 9 indicates that 63.82% of SPV is contributed by its shocks and that REC contributes by innovative shock for SPV with 5.76% in the long run. The contributions of PCD and LLD to SPV are minimal: 2.82% and 2.76%, respectively. Also, it was noted that the shock of SPV by itself estimates the largest percentages of forecast error in the short run and long run compared to other selected financial development indicators of forecast error. Nearly 8641% of PCD was significantly contributed by its innovative shock. The innovative shocks of REC, SPV, and LLD were enhanced in PCD by 4.32%, 4.37%, and 4.90%, respectively. The contributions of REC and PCD were negligible in LLD, estimated at 6.15% and 6.80%, respectively, while the innovative shocks of SPV contributed 63.82% to LLD and 13.84% of LLD was contributed by its innovative shock. This portion of the empirical proof resounds with the findings of [51,88,89].

### 4.6. Impulse Response Function Analyses

Finally, the impulse response function analyses were analyzed to illustrate the response in one variable due to shocks originating from other variables. Figure 3 plots the dynamic impact of one standard deviation of financial development indicator shocks on Saudi Arabia's renewable energy consumption over a horizon of 10 years. In Figure 3, the steady blue line symbolizes the impulse response of one variable (for instance, REC) to a one-standard-deviation shock to a different variable (for instance, PCD), whereas the dashed lines symbolize the upper and lower bounds of the 95% confidence intervals. Notably, REC shows positive responses to PCD, LLD, and SPV shock in the short run. Also, it can be observed that the impulse response function (IRF) is close to the zero line for the response of PCD to REC and LLD, which means that the system being analyzed does not respond to unexpected shocks or impulses in PCD. The author of [90] found that intraday volatility shocks as responses to macro-economic news result in elevated information asymmetry that causes adversity in price discovery.

**Table 9.** Forecast-error variance decomposition for the selected variables.

| Period | FEVD for LogREC | | | | FEVD for LogSPV | | | |
|---|---|---|---|---|---|---|---|---|
| | LogREC | LogSPV | LogPCD | LogLLD | LogREC | LogSPV | LogPCD | LogLLD |
| 1 | 1 | 0 | 0 | 0 | 0.013642 | 0.986358 | 0 | 0 |
| 2 | 0.924296 | 0.000498 | 0.07398 | 0.001227 | 0.040248 | 0.887699 | 0.003523 | 0.06853 |
| 3 | 0.863166 | 0.046957 | 0.070524 | 0.019353 | 0.037419 | 0.775254 | 0.002723 | 0.184604 |
| 4 | 0.785503 | 0.119789 | 0.076491 | 0.018217 | 0.034481 | 0.713203 | 0.006573 | 0.245744 |

**Table 9.** *Cont.*

| Period | FEVD for LogREC | | | | FEVD for LogSPV | | | |
| --- | --- | --- | --- | --- | --- | --- | --- | --- |
| | LogREC | LogSPV | LogPCD | LogLLD | LogREC | LogSPV | LogPCD | LogLLD |
| 5 | 0.738403 | 0.143864 | 0.085208 | 0.032526 | 0.04195 | 0.679115 | 0.015031 | 0.263905 |
| 6 | 0.707133 | 0.146983 | 0.092498 | 0.053386 | 0.048467 | 0.657358 | 0.021901 | 0.272274 |
| 7 | 0.68696 | 0.149686 | 0.103244 | 0.06011 | 0.05229 | 0.644544 | 0.025787 | 0.277378 |
| 8 | 0.672218 | 0.151272 | 0.115632 | 0.060878 | 0.055068 | 0.639355 | 0.027628 | 0.27795 |
| 9 | 0.662303 | 0.149972 | 0.126397 | 0.061328 | 0.056946 | 0.638118 | 0.028259 | 0.276677 |
| 10 | 0.655949 | 0.148189 | 0.134397 | 0.061465 | 0.057613 | 0.638201 | 0.028275 | 0.275911 |
| Period | FEVD for LogPCD | | | | FEVD for LogLLD | | | |
| | LogREC | LogSPV | LogPCD | LogLLD | LogREC | LogSPV | LogPCD | LogLLD |
| 1 | 0.075479 | 0.174743 | 0.749778 | 0 | 0.037532 | 0.1937 | 0.396734 | 0.372034 |
| 2 | 0.061696 | 0.1058 | 0.81364 | 0.018863 | 0.04636 | 0.129101 | 0.566059 | 0.258481 |
| 3 | 0.054756 | 0.081188 | 0.819653 | 0.044403 | 0.05217 | 0.163554 | 0.580642 | 0.203635 |
| 4 | 0.050658 | 0.06896 | 0.826832 | 0.05355 | 0.055497 | 0.16417 | 0.600129 | 0.180204 |
| 5 | 0.047545 | 0.060771 | 0.837663 | 0.054021 | 0.056583 | 0.151216 | 0.624903 | 0.167298 |
| 6 | 0.04553 | 0.054911 | 0.846448 | 0.053112 | 0.058388 | 0.140951 | 0.644608 | 0.156054 |
| 7 | 0.044506 | 0.050803 | 0.852323 | 0.052369 | 0.060864 | 0.133082 | 0.658039 | 0.148015 |
| 8 | 0.043919 | 0.047892 | 0.856767 | 0.051423 | 0.062063 | 0.127266 | 0.667348 | 0.143324 |
| 9 | 0.043479 | 0.045605 | 0.860704 | 0.050212 | 0.061966 | 0.123196 | 0.674501 | 0.140337 |
| 10 | 0.043211 | 0.043715 | 0.864053 | 0.049021 | 0.061462 | 0.120215 | 0.67994 | 0.138383 |

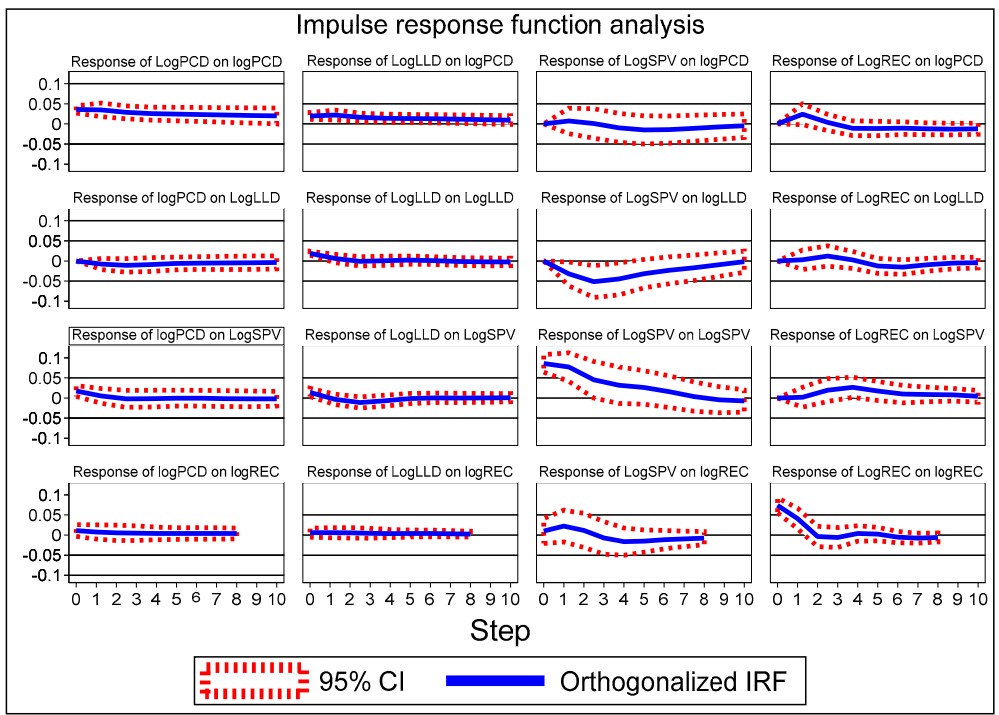

**Figure 3.** Response to generalized one SD innovations. Source: author's design (2023).

## 5. Conclusions and Policy Implications

During the last decades, the government of Saudi Arabia has adopted the reduction in fossil-fuel subsidies policy as a financial motivation for supporting both the production

and consumption of fossil fuels. The country launched the National Renewable Energy Program (NREP) plans to develop renewable energy projects and is working on developing the renewable energy sector through the partnership of public and private investment sectors. Therefore, this paper aims to explore the influence and shocks of Saudi's financial development indicators on renewable energy consumption and to establish the direction of causality between financial development indicators and renewable energy consumption.

This study uses the total renewable energy consumption (in TJ) as a proxy of sustainable development indicators; thus, three other proxies of financial development indicators are incorporated in the study: stock price volatility, private credit by deposit money banks to GDP (in %), and liquid liabilities to GDP (in %). The study covers the annual data period of 1990–2021 and applies some quantitative methodologies, including the Basic Vector Autoregressive model (VAR), the Granger causality test, the forecast-error variance decomposition (FEVD) test, and the impulse response function (IRF) test.

The empirical results of this study revealed dissimilar findings respecting the normality tests for the selected variables, so all variables were transformed to logarithm form. Also, the selected variables became stationary after their first differences. The results show that significant structural breaks and single dates were spotted in renewable energy consumption and financial development indicator variables.

The VAR results showed that, in the short run, stock price volatility and private credit significantly positively influence REC. Private credit impacts stock price volatility, while liquid liabilities exert a negative impact on stock price volatility. Likewise, private credit has a significantly positive influence on its changes and liquid liabilities.

The results from the asymmetric Granger causality test propose significant causality running from stock price volatility and private credit to REC. The results also show positive significant causality running from private credit and liquid liabilities to stock price volatility. The feedback of the hypotheses assumes that unidirectional Granger causality runs from stock price volatility and private credit to REC and that bi-directional Granger causality runs between stock price volatility and private credit. From our Granger findings, we concluded that the results of causality between REC and financial development indicators were conflicting.

Further, the results of FEVD revealed that more than half the percentage of the variation in REC was caused by itself, while liquid liabilities caused increasing variation in REC over 10 years. At the same time, stock price volatility and private credit caused decreasing variation in forecast error in REC over 10 years. This indicates that REC is shocked by itself, with the largest percentages of forecast error compared to the selected financial development indicators of forecast error throughout the 10 years. The empirical evidence indicates that stock price volatility is contributed by its shocks and that REC contributes to price volatility shock. The contributions of private credit and liquid liabilities to stock price volatility are minimal. Also, it was noted that the shock of price volatility by itself estimates the largest percentages of forecast error compared to other selected financial development indicators of forecast error throughout the 10 years. We also found that REC and stock price volatility contribute to innovation shocks of private credit. A fairly large portion of private credit is significantly contributed by its innovative shock. The contribution of REC private credit is negligible in liquid liabilities shocks, while the innovative shocks of price volatility contribute to liquid liabilities change. Also, a small portion of liquid liabilities is contributed by its innovative shocks.

The IRF results showed that REC is a positive response to shock on private credit, liquid liabilities, and stock price volatility. Also, it was noted that the impulse response function (IRF) is close to the zero line for the response of private credit to REC and liquid liabilities, which means that the system being analyzed does not respond to an unexpected shock (change) or "impulse" in private credit.

The policy implications of renewable energy consumption and financial development indicators are vital. Authorities can encourage investment in renewable energy consumption by providing financial incentives and motivations, such as tax reductions and subsidies;

facilitating access to financing for renewable energy projects; and establishing frameworks that will support the development of renewable energy markets. Financial policies for enhancing innovation in the renewable energy sector are significant for offering support and funds for research and supporting the development of new technologies. Additionally, the government can foster national partnerships between investors (financial institutions and companies), policymakers, and industry stakeholders, besides attracting international cooperation that can assist in accelerating the transition to a low-carbon economy and encouraging sustainable economic growth.

Further studies are suggested by employing different determinants of financial development indicators, such as non-bank financial institutions' assets to GDP, bank deposits to GDP (%), state-owned enterprises to GDP (%), and the cost of capital, including interest rates and lending terms, which can influence the affordability and attractiveness of renewable energy investments. Also, incorporating demographic characteristics, such as income level, household size, and geographical location, can affect the decision to adopt renewable energy technologies to reduce costs and environmental impact. In addition, introducing population growth in the REC function will be highly recommended for forming the renewable energy demand in Saudi Arabia. Furthermore, a regional comparative analysis of financial development indicators in neighboring Middle Eastern countries, specifically GCC countries, could offer valuable insights and comparative perspectives, and a micro-level analysis of specific sectors within renewable energy, such as solar, wind, and others, could provide actionable insights for policymakers. To obtain a comprehensive understanding of the effect of financial development indicators and renewable energy consumption policies in Saudi Arabia, it is fundamental to delve deeper into their societal implications. We suggested exploring how these advancements influence the lives of the common Saudi citizen, with a particular focus on job creation, cost savings, and overall quality of life. The study was often limited by the unavailability of micro-level sector data, specifically data related to solar and wind energy. Additionally, external factors such as global market dynamics can also pose limitations in conducting a comprehensive study.

**Supplementary Materials:** The following supporting information can be downloaded at: https://www.mdpi.com/article/10.3390/su152216004/s1.

**Funding:** This research was funded by the Deanship of Scientific Research, King Faisal University, Al-Ahsa, Saudi Arabia, under the grant contract, KFU Research Winter, Grant No.5048.

**Institutional Review Board Statement:** Not applicable.

**Informed Consent Statement:** Not applicable.

**Data Availability Statement:** Data are contained within the article and Supplementary Materials.

**Conflicts of Interest:** The author declares no conflict of interest.

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
