# Peer review of "Impact of Financial Development Shocks on Renewable Energy Consumption in Saudi Arabia"

_sustainability, doi:10.3390/su152216004_

Round 1
Reviewer 1 Report
Comments and Suggestions for Authors
I like to topic very much and it is very interesting and meaningful. The overall structure is ok and clear. I think the english writing is good to understand.
However, the last two paragraphs on page 2 are very strange. Why is there a number at the beginning of the paragraph? And also the second paragraph on page 3. Please check is it the right way?
The introduction part is too long, so can you make it shorter?
what is the specific research contribution and please describe in details. what is the innnovation points of your paper and could you please talk in further? could you please write more about the limitations of further research in the future in the last part of your paper?Author Response
Thank you very much for taking the time to review this manuscript. Thanks for all the suggestions and spotting points. We responded to all suggestions. Please find the detailed responses below and the corresponding revisions/corrections highlighted/in track changes in the re-submitted files. Comments 1: However, the last two paragraphs on page 2 are very strange. Why is there a number at the beginning of the paragraph? And also, the second paragraph on page 3. Please check is it the right way? |
Response 1: Thank you for pointing this out. We agree with this comment. Therefore, We corrected it and we rewrote the sentences accordingly in red color. In lines 179, 181and 210. |
Comments 2: The introduction part is too long, so can you make it shorter? |
Response 2: Agree, The introduction included the literature review and the status of Saudi Arabia of renewable energy and financial development. Accordingly, we break down the introduction to emphasize this point. In lines from 69-261. |
Comments 3: what is the specific research contribution and please describe it in detail. What is the innovation points of your paper and could you please talk in further? could you please write more about the limitations of further research in the future in the last part of your paper? |
Response 3: We agree with these comments. The contribution of the paper was improved. A new point of innovation/novel was added in lines 141-144, also limitations of further research were added in the last paragraph of the conclusion in red color.
|
4. Response to Comments on the Quality of English Language |
Point 1: No issues detected. |
Response 1: No issues detected. |

Reviewer 2 Report
Comments and Suggestions for Authors
My comments are as follows to improve the article further:
1. It would be helpful to briefly explain the current state of renewable energy consumption in Saudi Arabia and why this issue is of particular importance in that context?
2. The contributions in the introduction sections needs to be strengthen further.
3. The research objective is somewhat vague. It states that the study aims to investigate the influence of financial development indicators on REC and to examine the causality between them. It would be beneficial to specify the exact questions the study seeks to answer or the hypotheses it aims to test.
4. In addition, the study mentions the use of specific methodologies (VAR, Granger causality test, FEVD, IRF), which is good. However, it would be helpful to briefly explain why these methods were chosen and what the significance of each is in this context.
5. The results of causality between REC and financial development indicators were conflicting," is quite obscure. It would be better to briefly explain the nature of this conflict or re-write specifically in the abstract part.
6. The Introduction section is too long, I recommend to breakdown the section and create and perform a "Literature review". In addition, so many studies are conducted on Renewable energy consumption and its impact on financial sector development", kindly incorporate those, and mention how your study is unique? what additional contributions you have made? Some of the studies more specifically done for MENA and KSA are as under, kindly incorporate them while highlighting and stressing your contributions:
https://www.sciencedirect.com/science/article/abs/pii/S136403211630822X
https://link.springer.com/article/10.1007/s11356-022-23867-z
https://link.springer.com/article/10.1007/s11356-020-10445-4
https://www.sciencedirect.com/science/article/abs/pii/S0301421504003672
https://link.springer.com/article/10.1007/s11356-020-08390-3
https://www.sciencedirect.com/science/article/pii/S1364032122006487
7. Moreover, the discussion section is lacking a clear flow and interpretation of results. The focus is on summarizing the results of the study, and agreeing or contradicting with the previous work. However, the focus should be on Interpretation of the results and explanation of their implications. Discuss the meaning of your findings and why they matter economically? and for enhancement and achieving sustainability? Avoid repeating the results; instead, provide context and analysis.
8. check for grammatical errors, typos, and improve the flow of writing.
9. Briefly state limitations of the study in the conclusion section once you summarize your results and mention future directions.
Comments on the Quality of English LanguageCheck for grammatical errors, typos, and there is a need to improve the flow of writing.
Author Response
Comments 1: It would be helpful to briefly explain the current state of renewable energy consumption in Saudi Arabia and why this issue is of particular importance in that context?
|
Response 1: Thank you for pointing this out. We agree with this comment. The current state of renewable energy in Saudi consumption is already in the introduction, so we separated it into section: “1.1 Financial development and renewable energy consumption in Saudi Arabia”, also some improvement was made in red color. In lines 69-133. |
Comments 2: The contributions in the introduction sections needs to be strengthen further. |
Response 2: Agree, the contribution of the paper was strengthened accordingly in red color. In lines 134-144. |
Comments 3: The research objective is somewhat vague. It states that the study aims to investigate the influence of financial development indicators on REC and to examine the causality between them. It would be beneficial to specify the exact questions the study seeks to answer or the hypotheses it aims to test. |
Response 3: Thanks for spotting this. Two significant questions were added before the aims of the study. In lines 128-131, to emphasize this point. |
Comments 4: In addition, the study mentions the use of specific methodologies (VAR, Granger causality test, FEVD, IRF), which is good. However, it would be helpful to briefly explain why these methods were chosen and what the significance of each is in this context |
Response 4: We agree with you. We briefly explained why we chose these methods and we added their overall significance in red color. In lines: 338-343; 382-385 and 411-414. |
Comments 5: The results of causality between REC and financial development indicators were conflicting," is quite obscure. It would be better to briefly explain the nature of this conflict or re-write specifically in the abstract part. |
Response 5: Agree. We explain the nature of this conflict in section 4.4 in red color. Also, we briefly added some results in the Abstract in red color. |
Comments 6: The Introduction section is too long, I recommend to breakdown the section and create and perform a "Literature review". In addition, so many studies are conducted on Renewable energy consumption and its impact on financial sector development", kindly incorporate those, and mention how your study is unique? what additional contributions you have made? Some of the studies more specifically done for MENA and KSA are as under, kindly incorporate them while highlighting and stressing your contributions: https://www.sciencedirect.com/science/article/abs/pii/S136403211630822X https://link.springer.com/article/10.1007/s11356-022-23867-z https://link.springer.com/article/10.1007/s11356-020-10445-4 https://www.sciencedirect.com/science/article/abs/pii/S0301421504003672 https://link.springer.com/article/10.1007/s11356-020-08390-3 https://www.sciencedirect.com/science/article/pii/S1364032122006487
|
Response 6: Yes, you are right. We break down the literature view from the introduction. Accordingly, we break down the introduction to emphasize this point. In lines from 69-261. Also, all suggested new references were cited: https://www.sciencedirect.com/science/article/abs/pii/S136403211630822X Response: Already was incorporated in red color in the reference part. https://link.springer.com/article/10.1007/s11356-022-23867-z Response: We incorporated it in red color in the reference part. https://link.springer.com/article/10.1007/s11356-020-10445-4 Response: We incorporated it in red color in the reference part. https://www.sciencedirect.com/science/article/abs/pii/S0301421504003672 Response: We incorporated it in red color in the reference part. https://link.springer.com/article/10.1007/s11356-020-08390-3 Response: We incorporated it in red color in the reference part. https://www.sciencedirect.com/science/article/pii/S1364032122006487 Response: We incorporated it in red color in the reference part. |
Comments 7: Moreover, the discussion section is lacking a clear flow and interpretation of results. The focus is on summarizing the results of the study, and agreeing or contradicting with the previous work. However, the focus should be on Interpretation of the results and explanation of their implications. Discuss the meaning of your findings and why they matter economically? and for enhancement and achieving sustainability? Avoid repeating the results; instead, provide context and analysis. |
Response 7: Agree. We provide some context and analysis of our findings in red color. In lines 468-471, 548-550,563-564,583-587,588-593, 599-600, 618, 620. |
Comments 8 : check for grammatical errors, typos, and improve the flow of writing. |
Response 8 : Grammatical errors were checked. |
Comments 9. Briefly state limitations of the study in the conclusion section once you summarize your results and mention future directions. |
Response 9: Thanks for pointing out this point. The limitations of this study were briefly stated. In lines 732-734. |
Comments 10: |
4. Response to Comments on the Quality of English Language |
Point 1: |
Response 1 Grammatical errors were checked. |
5. Additional clarifications |
[Here, mention any other clarifications you would like to provide to the journal editor/reviewer.] |

Reviewer 3 Report
Comments and Suggestions for Authors
This paper examines the track of causality between financial development indicators and renewable energy consumption through various methods, such as VAR, Granger causality test, and FEVD. The research has a novel perspective, rich research methods, practical relevance in the topic selection, and coherent writing. However, the following issues still exist:
1. The sample size in Table 5, which covers the period from 1994 to 2021, is only 28, which cannot guarantee that the distribution of sample means approximates a normal distribution.
2. The research focuses solely on Saudi Arabia as the study subject, with a small sample size of 32. This limits the persuasiveness of the research findings. Whether it is possible to extend the study to include cities within Saudi Arabia and construct panel data to examine the impact of financial shocks on renewable energy?
3. This paper provides rich research findings through VAR, Granger causality tests, FEVD, and IRF. It would be better if explanations for the reasons behind the relevant conclusions in the empirical analysis are included.
4. The paper links financial shocks to renewable energy based on the description of the Saudi government's reduction of fossil fuel subsidies and the initiation of the National Renewable Energy Program (NREP) in 2019. While this approach has strong empirical support, it lacks theoretical underpinnings. It would be better if enhance the theoretical and empirical analysis on the mechanisms through which financial shocks affect renewable energy.
5. This paper relies on the traditional VAR model for the research, which has certain limitations in the methodology. It is recommended to employ the TVP-VAR model (Time Varying Parameter-Stochastic Volatility-Vector Auto Regression), also known as the time-varying parameter stochastic volatility vector autoregressive model. Compared to the traditional VAR model, this approach does not assume homoscedasticity, which is more in line with real-world situations. Additionally, it utilizes the more advanced MCMC algorithm for estimation.
Comments on the Quality of English LanguageMinor editing of English language required
Author Response
Thank you very much for taking the time to review this manuscript. Thanks for all the suggestions and spotting points. We responded to all suggestions. Please find the detailed responses below and the corresponding revisions/corrections highlighted/in track changes in the re-submitted files.
Comments 1: The sample size in Table 5, which covers the period from 1994 to 2021, is only 28, which cannot guarantee that the distribution of sample means approximates a normal distribution. |
Response 1: Thank you for pointing this out. We agree with this comment. Yes, you are right. For that, all variables are transferred in logarithms form. |
Comments 2: The research focuses solely on Saudi Arabia as the study subject, with a small sample size of 32. This limits the persuasiveness of the research findings. Whether it is possible to extend the study to include cities within Saudi Arabia and construct panel data to examine the impact of financial shocks on renewable energy? |
Response 2: Agree, Yes, you are right. However, there is no data available for the cities within Saudi Arabia. E.g. Saudi Arabia also has significant potential for solar and wind energy development, particularly in coastal regions. The government has initiated wind energy projects and conducted feasibility studies to harness this resource. However, wind energy is still in its early stages, and the contribution of wind power to the country's energy consumption is relatively limited. We added this limitation briefly in the conclusion part. |
Comments 3: This paper provides rich research findings through VAR, Granger causality tests, FEVD, and IRF. It would be better if explanations for the reasons behind the relevant conclusions in the empirical analysis are included. |
Response 3: Thanks for spotting this. We briefly explained why we have chosen these methods and we add their overall significance in red color. In lines: 338-343; 382-385 and 411-414. |
Comments 4: The paper links financial shocks to renewable energy based on the description of the Saudi government's reduction of fossil fuel subsidies and the initiation of the National Renewable Energy Program (NREP) in 2019. While this approach has strong empirical support, it lacks theoretical underpinnings. It would be better if enhanced the theoretical and empirical analysis on the mechanisms through which financial shocks affect renewable energy. |
Response 4: Thanks for pointing out this comment. Theoretical underpinnings were already added in the literature review. Also, new references were added in red color in the reference part. |
Comments 5: This paper relies on the traditional VAR model for the research, which has certain limitations in the methodology. It is recommended to employ the TVP-VAR model (Time Varying Parameter-Stochastic Volatility-Vector Auto Regression), also known as the time-varying parameter stochastic volatility vector autoregressive model. Compared to the traditional VAR model, this approach does not assume homoscedasticity, which is more in line with real-world situations. Additionally, it utilizes the more advanced MCMC algorithm for estimation. |
Response 5: Yes you are right. The TVP-VAR model is better and typically employs Bayesian methods to estimate the time-varying parameters. These methods allow for the estimation of the parameters' posterior distributions over time, providing information about their evolving values and uncertainty. However, according to the nature of our objective, we used the Basic VAR model because it assumes that the relationships captured by the coefficients are stable and do not vary with different economic conditions or periods. Also, our variables are small. |
Comments 6: |
4. Response to Comments on the Quality of English Language |
Point 1: |
Response 1 |
5. Additional clarifications |
[Here, mention any other clarifications you would like to provide to the journal editor/reviewer.] |

Reviewer 4 Report
Comments and Suggestions for Authors
These are major revision required for article to be published.
Comments of the reviewer
1. The abstract should be revised on the basis of what has been done and what’s new in this research?
2. There is no comparative analysis made which can compare the efficiency of this study with similar type of studies.
3. What is novelty of the paper?
4. How the mathematical model was validated or verified. Give complete validation results.
5. Add more detailed introduction with the help of latest article published in the field of study.
6. There are many English Grammer mistakes which should be improved.
7. The work offers insightful suggestions for future research, like taking demographic characteristics and other determinants of financial development indicators into account, although it would be helpful to go into more detail on how these factors can interact to affect REC in Saudi Arabia. Explain all the factors and how they affect REC.
8. The data in the report only extends to 2021, therefore it may not accurately reflect current changes to Saudi Arabia's energy environment. Investments in and policies related to renewable energy can change quickly, therefore utilizing more recent data would improve the study's relevance and accuracy.
9. The analysis ignores other important factors that affect the adoption of renewable energy in favour of concentrating largely on financial development metrics as REC drivers. There is a cursory reference but little exploration of factors including public awareness, government regulations, and technology improvements. A more thorough examination of these factors might offer a more complex comprehension of the dynamics in operation.
10. Regarding the causal relationship between financial development metrics and REC, the article presents contradictory findings. Even while it is common for complicated economic linkages to have inconsistent outcomes, the research ought to offer a more thorough analysis and plausible justifications for these differences. The reader is left wondering how solid the results are.
11. Why Granger causality approach was chosen and how it is better as compared to other techniques for the same purpose.
12. Table 5 need further explanation on how the criteria was set?
Comments on the Quality of English LanguageMinor changes required as far quality of English is concerned.
Author Response
Thank you very much for taking the time to review this manuscript. Thanks for all the suggestions and spotting points. We responded to all suggestions. Please find the detailed responses below and the corresponding revisions/corrections highlighted/in track changes in the re-submitted files.
Comments 1: 1. The abstract should be revised on the basis of what has been done and what’s new in this research? |
The Abstract was revised. In lines 21-22, 23-24 and 26-28. |
Comments 2: There is no comparative analysis made which can compare the efficiency of this study with similar type of studies. |
Response 2: Thanks for pointing out this. We compared our study with other studies in the last paragraph of the review, in lines 257-262. |
Comments 3: What is novelty of the paper? |
Response 3: Thanks for pointing out this. Novelty of the study was added. In lines 141-143. |
Comments 4: How the mathematical model was validated or verified. Give complete validation results. |
Response 4: We agree with you. We examined several tests for verifying the VAR by test for e.g., stationery, lag order, Lagrange Multiplier (LM), and forecasting tests. For e.g. in lines 450-453. |
Comments 5: Add more detailed introduction with the help of latest article published in the field of study. |
Response 5: Agree. We added some studies specifically conducted in Saudi Arabia in red color in the references part. |
Comments 6: There are many English Grammer mistakes which should be improved. |
Response 6: Grammar mistakes were improved accordingly. |
Comments 7: The work offers insightful suggestions for future research, like taking demographic characteristics and other determinants of financial development indicators into account, although it would be helpful to go into more detail on how these factors can interact to affect REC in Saudi Arabia. Explain all the factors and how they affect REC. |
Response 7: Yes, we agree. We added some suggested indicators in the conclusion part. In the last paragraph with red color. |
Comments 8: The data in the report only extends to 2021, therefore it may not accurately reflect current changes to Saudi Arabia's energy environment. Investments in and policies related to renewable energy can change quickly, therefore utilizing more recent data would improve the study's relevance and accuracy |
Response 8: Yes, we agree. But the data is available till this year 2021. |
Comments 9. The analysis ignores other important factors that affect the adoption of renewable energy in favour of concentrating largely on financial development metrics as REC drivers. There is a cursory reference but little exploration of factors including public awareness, government regulations, and technology improvements. A more thorough examination of these factors might offer a more complex comprehension of the dynamics in operation. |
Response 9: Yes, we agree. However, the data limitation is the main reason for not involving other factors in the model. |
Comments 10: Regarding the causal relationship between financial development metrics and REC, the article presents contradictory findings. Even while it is common for complicated economic linkages to have inconsistent outcomes, the research ought to offer a more thorough analysis and plausible justifications for these differences. The reader is left wondering how solid the results are. |
Response 10: Yes, there are contradictory findings,s in particular the causality test, The plausible justifications of contradictory findings were added accordingly in red color. In lines:588-593. |
Comment 11: Why Granger causality approach was chosen and how it is better as compared to other techniques for the same purpose. |
Response 11: Thanks for pointing out these comments. We justified why the Granger causality approach was chosen. Also, the overall justifications for why we chose these econometrics models were added in lines: 338-343; 382-385, and 411-414. |
Comment 12: Table 5 need further explanation on how the criteria was set? |
Response 12: Agree. Further explanation was added in lines: 517-519. |
4. Response to Comments on the Quality of English Language |
Point 1: |
Response 1 |
5. Additional clarifications |
[Here, mention any other clarifications you would like to provide to the journal editor/reviewer.] |

Reviewer 5 Report
Comments and Suggestions for Authors
The last few decades have witnessed a significant shift in the approach of governments worldwide towards sustainable development. This paper uniquely positions itself at the intersection of Saudi Arabia's efforts in reducing fossil fuel subsidies and the unfolding dynamics of renewable energy consumption (REC). It delves into the intricacies of the National Renewable Energy Program (NREP) and seeks to understand the influence of the Kingdom's financial development indicators on its renewable energy journey.
A commendable aspect of this research is its rigorous methodological approach. By selecting total renewable energy consumption as a proxy for sustainable development, the study brings into focus other financial development indicators such as stock price volatility, private credit to GDP ratio, and liquid liabilities to GDP ratio. The chosen period (1990-2021) ensures a comprehensive analysis, and the application of quantitative methodologies like VAR, Granger causality test, FEVD, and IRF adds depth and robustness.
The findings of the paper are multifaceted. Firstly, it highlights the significance of structural breaks in renewable energy consumption and associated financial development indicators. Secondly, the VAR results elucidate the short-term influences of stock price volatility and private credit on REC. The Granger-causality test further provides insights into the direction of causality between these indicators and REC, albeit with some conflicting results.
Furthermore, the FEVD results shed light on the contribution of various factors to the variation in REC over a decade. It is intriguing to note that REC itself is a significant influencer. The IRF results are also illuminating, showcasing the reactions of REC to shocks in various financial indicators.
The paper seamlessly transitions from empirical findings to policy implications, emphasizing the importance of financial incentives, facilitative frameworks, and collaborative partnerships in accelerating renewable energy adoption. These recommendations are highly actionable and aptly address the challenges and opportunities present in the Saudi context.
To further refine this already comprehensive study, a few recommendations are to include in discussion (nbo new deep investigation needed):
->regional Comparative Analysis: Investigating how neighboring Middle Eastern countries, with similar economic structures and energy matrices, respond to financial development indicators could offer comparative insights.
->Micro-level Analysis: While macro indicators provide a broad perspective, a granular look into specific sectors within renewable energy (like solar, wind, etc.) can provide actionable insights for policymakers.
->Societal Implications: A deeper dive into how these financial developments and renewable energy policies impact the common Saudi citizen, in terms of job creation, cost savings, and overall quality of life, can provide a more holistic picture.
In conclusion, this paper provides a meticulous analysis of Saudi Arabia's renewable energy trajectory in the context of its financial developments. The insights gleaned are invaluable not only for the Kingdom but also for other nations grappling with similar challenges and seeking a sustainable future. With the suggested extensions, the study can provide even more enriched insights into the intricate dance of finance and renewable energy.
Author Response
Thank you very much for taking the time to review this manuscript. Thanks for all the suggestions and spotting points. We responded to all suggestions. Please find the detailed responses below and the corresponding revisions/corrections highlighted/in track changes in the re-submitted files.
Comments 1: The last few decades have witnessed a significant shift in the approach of governments worldwide towards sustainable development. This paper uniquely positions itself at the intersection of Saudi Arabia's efforts in reducing fossil fuel subsidies and the unfolding dynamics of renewable energy consumption (REC). It delves into the intricacies of the National Renewable Energy Program (NREP) and seeks to understand the influence of the Kingdom's financial development indicators on its renewable energy journey. |
Response 1: Thanks for your comments. |
Comments 2: A commendable aspect of this research is its rigorous methodological approach. By selecting total renewable energy consumption as a proxy for sustainable development, the study brings into focus other financial development indicators such as stock price volatility, private credit to GDP ratio, and liquid liabilities to GDP ratio. The chosen period (1990-2021) ensures a comprehensive analysis, and the application of quantitative methodologies like VAR, Granger causality test, FEVD, and IRF adds depth and robustness. |
Response 2: Thanks for your comments. |
Comments 3: The findings of the paper are multifaceted. Firstly, it highlights the significance of structural breaks in renewable energy consumption and associated financial development indicators. Secondly, the VAR results elucidate the short-term influences of stock price volatility and private credit on REC. The Granger-causality test further provides insights into the direction of causality between these indicators and REC, albeit with some conflicting results. |
Response 3: Thanks for your comments. |
Comments 4: Furthermore, the FEVD results shed light on the contribution of various factors to the variation in REC over a decade. It is intriguing to note that REC itself is a significant influencer. The IRF results are also illuminating, showcasing the reactions of REC to shocks in various financial indicators. |
Response 4: Thanks for your comments |
Comments 5: The paper seamlessly transitions from empirical findings to policy implications, emphasizing the importance of financial incentives, facilitative frameworks, and collaborative partnerships in accelerating renewable energy adoption. These recommendations are highly actionable and aptly address the challenges and opportunities present in the Saudi context. |
Response 5: Thanks for your comments |
Comments 6: To further refine this already comprehensive study, a few recommendations are to include in discussion (nbo new deep investigation needed): Regional Comparative Analysis: Investigating how neighboring Middle Eastern countries, with similar economic structures and energy matrices, respond to financial development indicators could offer comparative insights. |
Response 6: Thank you for pointing out this. We added this recommended in the conclusion part. In lines: 723-725. |
Comments 7: Micro-level Analysis: While macro indicators provide a broad perspective, a granular look into specific sectors within renewable energy (like solar, wind, etc.) can provide actionable insights for policymakers. |
Response 7: Thank you for pointing out this. we added this recommended in the conclusion part, in lines: 725-727. |
Comments 8: Societal Implications: A deeper dive into how these financial developments and renewable energy policies impact the common Saudi citizen, in terms of job creation, cost savings, and overall quality of life, can provide a more holistic picture. |
Response 8: Thank you for pointing out this. We added this recommended in the conclusion part. In lines 727-732. |
Comments 9: In conclusion, this paper provides a meticulous analysis of Saudi Arabia's renewable energy trajectory in the context of its financial developments. The insights gleaned are invaluable not only for the Kingdom but also for other nations grappling with similar challenges and seeking a sustainable future. With the suggested extensions, the study can provide even more enriched insights into the intricate dance of finance and renewable energy. |
Response 9: Thanks for your comments |
4. Response to Comments on the Quality of English Language |
Point 1: |
Response 1: (in red) |
5. Additional clarifications |
[Here, mention any other clarifications you would like to provide to the journal editor/reviewer.] |

Reviewer 6 Report
Comments and Suggestions for Authors
1. Strong econometric paper, but intuition can be enhanced
2. Heteroscedasticity reference needs to be fixed. Not multicollinearity, rather non-constant variance of error term
3. Unit root and eigenvector mapping to eigenvalue modulus test needs to be developed
4. Pls refer to pdf annotations for minor typos and updates.

Author Response
Thank you very much for taking the time to review this manuscript. Thanks for all the suggestions and spotting points. We responded to all suggestions. Please find the detailed responses below and the corresponding revisions/corrections highlighted/in track changes in the re-submitted files.
Comments 1: Strong econometric paper, but intuition can be enhanced. Heteroscedasticity reference needs to be fixed. Not multicollinearity, rather non-constant variance of error term. |
Response 1: Thanks for pointing out this. The Heteroscedasticity reference is fixed in line 451. |
Comments 2: Unit root and eigenvector mapping to eigenvalue modulus test needs to be developed. |
Response 2: Thanks for your comments. We developed the unit root results in red color and the eigenvalue mapping and modules. In lines: 468-471, in the red color in Table 7 and Fig. 2 |
Comments 3: Pls refer to pdf annotations for minor typos and updates |
Response 3: Thanks for your comments. We referred, and all corrections were done. 1. We separated the literature review from the introduction; accordingly, we break down the introduction to emphasize this point. In lines from 69-261. 2. We transfer sentences beginning with: Another study evaluates the relationship between REC and financial development employing panel nonlinear Autoregressive Distributed Lag (ARDL)……. in lines: 237-238. 3. We change 360 days to one year. In Table 1 with red color. 4. We change testes to tests. In line 299. 5. We express all terms t, T, and D Y, after equations (9), and we transfer them in the after equation (6). 6. Response to the comments: not all variables are defined at least one: all variables must be defined because their values for accept or reject hypotheses have differed. 7. We corrected the word TB to TBD in equations (10 and 11). 8. In the past section 2 was Materials and Methods now it's Section 3, because we added a new section of literature review. So, no change of the word section 3 to section 2. 9. We deleted the words: unit root test in line 487. 10. We change the sentence: show a serial correlation of the series: with “shows that the variance error is not constant” in red color, in line 497. 11. We used log not Ln, because Logarithms are particularly useful when dealing with percentage changes. So some of our data are percentages (e.g. PCD). Also, we follow the researchers for the chosen log for similar studies. e.g., https://doi.org/10.1016/j.erss.2020.101537
|
4. Response to Comments on the Quality of English Language |
Point 1: No issues detected |
Response 1: No issues detected |
5. Additional clarifications |
11. Some comments with handwriting I can’t read them properly. |

Reviewer 7 Report
Comments and Suggestions for Authors
This paper investigates the influence and shocks of Saudi Arabia’s financial development indicators on renewable energy consumption.
Though the paper is fairly well written, it is not an easy read. The author has used several statistical techniques and the transition from one section to another is not very smooth making it a little difficult to read. In addition, all the statistical techniques that the author has used are well-established techniques and it is not necessary to explain these techniques using a lot of formulas. Nevertheless, the statistical techniques that are used in the paper are appropriate for a study of this nature.
Though the author has discussed the results of the paper well, it is necessary to make some changes to some of the tables.
1. In Table 1 under the materials and methods section (Section 2), variable explanation could be included as a note to the table instead of including it in a separate column in the table. It is also not clear what TJ stands for when REC variable is explained.
2. In Table 2, the results indicate that all the variables used in the model suffer from serial correlation. It is not clear if the logarithmic transformation resolves this problem.
3. In Table 4, it is not necessary to present both the z-statistic as well as the standard errors of the coefficients. One can easily derive the z-statistic using the estimated coefficients and their standard errors.
4. In Table 7, the lower and upper limits of eigenvalues could have been presented instead of using + or – values.
5. In Figure 3, since the focus of the paper is to investigate the influence and shocks of Saudi Arabia’s financial development indicators on renewable energy consumption, only the last four impulse response functions could have presented.
The conclusions adequately tie together the other elements of the paper. To some extent, the paper has expressed its case, measured against the technical language of the field and the expected knowledge of the journal's readership.

Comments on the Quality of English LanguageThe quality of English language is good.
Author Response
Thank you very much for taking the time to review this manuscript. Thanks for all the suggestions and spotting points. We responded to all suggestions. Please find the detailed responses below and the corresponding revisions/corrections highlighted/in track changes in the re-submitted files.
Comments 1: This paper investigates the influence and shocks of Saudi Arabia’s financial development indicators on renewable energy consumption. Though the paper is fairly well written, it is not an easy read. The author has used several statistical techniques and the transition from one section to another is not very smooth making it a little difficult to read. In addition, all the statistical techniques that the author has used are well-established techniques and it is not necessary to explain these techniques using a lot of formulas. Nevertheless, the statistical techniques that are used in the paper are appropriate for a study of this nature. Though the author has discussed the results of the paper well, it is necessary to make some changes to some of the tables. In Table 1 under the materials and methods section (Section 2), variable explanation could be included as a note to the table instead of including it in a separate column in the table. It is also not clear what TJ stands for when REC variable is explained. |
Response 1: Thanks for your comments. we included the variable explanation as a note in the table (1) and we deleted the column. TJ is stands for terajoules. We explained this in the section (3.1), data and descriptive statistics section) with red color. |
Comments 2: In Table 2, the results indicate that all the variables used in the model suffer from serial correlation. It is not clear if the logarithmic transformation resolves this problem. |
Response 2: Thanks for your comments. This was mentioned in the last paragraph before the table 2. In line 481. |
Comments 3: In Table 4, it is not necessary to present both the z-statistic as well as the standard errors of the coefficients. One can easily derive the z-statistic using the estimated coefficients and their standard errors. |
Response 3: Thanks for your comments. The Z-statistics were deleted from the table 4. |
Comments 4: In Table 7, the lower and upper limits of eigenvalues could have been presented instead of using + or – values |
Response 4: Thanks for pointing out this. The "+" and "-" signs indicate the sign of the imaginary part. We clarified them as the note in Table 7. |
Comments 5: In Figure 3, since the focus of the paper is to investigate the influence and shocks of Saudi Arabia’s financial development indicators on renewable energy consumption, only the last four impulse response functions could have presented. |
Response 5: Yes, you are right. We presented only the responses of REC in the text. However, we displayed other responses in the figure to help further studies/ researchers observe the comparison of interaction responses among the variables for consideration in further studies. |
Comments 6: The conclusions adequately tie together the other elements of the paper. To some extent, the paper has expressed its case, measured against the technical language of the field and the expected knowledge of the journal's readership. |
Response 6: Thank you for your comments. |
4. Response to Comments on the Quality of English Language |
Point 1: |
Response 1: (in red) |
5. Additional clarifications |
[Here, mention any other clarifications you would like to provide to the journal editor/reviewer.] |

Round 2
Reviewer 2 Report
Comments and Suggestions for Authors
The paper can be accepted, no further comments.
Author Response
Thanks for your efforts and valuable time for reviewing our article.
Response: No extra comment has been sent.
Reviewer 4 Report
Comments and Suggestions for Authors
All the changes has been made now and its fine to be published.
Author Response

(The authors gave the same response as above.)

Reviewer 5 Report
Comments and Suggestions for Authors
Paper can be accepted now
Author Response

(The authors gave the same response as above.)

Reviewer 6 Report
Comments and Suggestions for Authors
Dear Author/s,
Thank you for reverting back and incorporating feedback on your mathematical notations and other details. It takes time at both ends to refine a paper. I had taken time and written out the Eigenvector formulation, however I see that you've made some minor changes to the eigenvalues and left it at that. That is fine...
To make your references span wider and increase the basis of your text a few references were written out by hand. Sorry they were not legible. Please find relevant text that can be introduced near the line numbers of the original version.
The following would increase the scope of your paper to a finance audience. These are topical suggestions and are not to be interpreted as a recommendation:
Line 448: "Volatility impacts option prices exponentially, hence the dilutive effect of in-the-money options on blended earnings can affect growth estimates adversely, making volatile companies harder to hold, this has been shown by (Goldsticker and Agrrawal, The Effects of Blending Primary and Diluted EPS Data (1999). Financial Analysts Journal, 1999)
Line 573: Jurdi (2020, Intraday Jumps, Liquidity and Macroeconomic News …Journal of risk and financial management ) finds that intraday volatility shocks as a response to macroeconomic news results in elevated information asymmetry that causes adversity in price discovery.
Thank you for incorporating the modeling ( Eigenvector and Heteroscedasticity) related feedback. I know it was work, but in the end the objective is to produce a best possible manuscript that has minimal obvious blemishes to a reader. Thank you.
Author Response
Thank you very much for taking the time to review this manuscript. Thanks for all the suggestions and spotting points. We responded to all suggestions. Please find the detailed responses below and the corresponding revisions/corrections highlighted/in track changes in the re-submitted revised (2) files.
Comments 1: Thank you for reverting back and incorporating feedback on your mathematical notations and other details. It takes time at both ends to refine a paper. I had taken time and written out the Eigenvector formulation, however I see that you've made some minor changes to the eigenvalues and left it at that. That is fine...
Response 1 Thanks for mentioning this gain. We added some extra explanation in lines: 543-548
Comments 2: To make your references span wider and increase the basis of your text a few references were written out by hand. Sorry they were not legible. Please find relevant text that can be introduced near the line numbers of the original version.
Response 2: Thanks for pointing out this point. We revised all the handwriting refences in red colour.
The following would increase the scope of your paper to a finance audience. These are topical suggestions and are not to be interpreted as a recommendation:
Comments 3:Line 448: "Volatility impacts option prices exponentially, hence the dilutive effect of in-the-money options on blended earnings can affect growth estimates adversely, making volatile companies harder to hold, this has been shown by (Goldsticker and Agrrawal, The Effects of Blending Primary and Diluted EPS Data (1999). Financial Analysts Journal, 1999)
Response 3: Thank for mentioned this, we added it accordingly. In lines: 500-502
Comments 4:Line 573: Jurdi (2020, Intraday Jumps, Liquidity and Macroeconomic News …Journal of risk and financial management ) finds that intraday volatility shocks as a response to macroeconomic news results in elevated information asymmetry that causes adversity in price discovery.
Response 4: Thank for mentioned this, we added it accordingly. In lines: 655-657
Comments 5:Thank you for incorporating the modeling ( Eigenvector and Heteroscedasticity) related feedback. I know it was work, but in the end the objective is to produce a best possible manuscript that has minimal obvious blemishes to a reader. Thank you.
Response 5: Thanks for your efforts and time for reviewing our article.
Reviewer 7 Report
Comments and Suggestions for Authors
The revised version of the manuscript shows some improvement over the original manuscript. The author has addressed all my concerns about the previous version of the manuscript.
Comments on the Quality of English LanguageThe quality of English language is good.
Author Response

(The authors gave the same response as above.)

Round 3
Reviewer 6 Report
Comments and Suggestions for Authors
Your introduction of the imaginary part of a complex number is very useful to understand how you've applied eigenvectors to your analysis. "Consequently, the corresponding eigenvectors associated with these eigenvalues do not experience rotation or oscillation but instead undergo scaling or stretching along a specific direction. Table 7" ; line 548
This will be an important addition to the literature on Financial shocks and Renewable energy. Perhaps this will be factored into the development of the Green Initiative NEOM / The Line as part of the SGI.